# Emerging climate signals in the Lena River catchment: a non-parametric statistical approach

Eric Pohl[1], Christophe Grenier[1], Mathieu Vrac[1], Masa Kageyama[1]

[1]Laboratoire des Sciences du Climat et de l'Environnement (LSCE/IPSL), UMR CEA-CNRS-UVSQ, Gif-sur-Yvette, 91120, France

*Correspondence to*: Eric Pohl (Eric.Pohl@lsce.ipsl.fr)

**Abstract.** Climate change has far-reaching implications in permafrost-underlain landscapes with respect to hydrology, ecosystems and the population's traditional livelihoods. In the Lena River catchment, eastern Siberia, changing climatic conditions and the associated impacts are already observed or expected. However, as climate change progresses the question remains as to how far we are along this track and when these changes will constitute a significant emergence from natural variability. Here we present an approach to investigate temperature and precipitation time series from observational records, reanalysis, and an ensemble of 65 climate model simulations forced by the RCP8.5 emission scenario. We developed a novel non-parametric statistical method to identify the time of emergence (ToE) of climate change signals, i.e. the time when a climate signal permanently exceeds its natural variability. The method is based on the Hellinger distance metric that measures the similarity of probability density functions (PDFs) roughly corresponding to their geometrical overlap. Natural variability is estimated as a PDF for the earliest period common to all datasets used in the study (1901-1921) and is then compared to PDFs of target periods with moving windows of 21 years at annual and seasonal scales. The method yields dissimilarities or emergence levels ranging from 0 to 100% and the direction of change as a continuous time series itself. First, we showcase the method's advantage over the Kolmogorov-Smirnov metric using a synthetic dataset that resembles signals observed in the utilized climate models. Then, we focus on the Lena River catchment, where significant environmental changes are already apparent. On average, emergence of temperature has a strong onset in the 1970s with a monotonic increase thereafter for validated reanalysis data. At the end of the reanalysis dataset (2004), temperature distributions have emerged by 50-60%. Climate model projections suggest the same evolution on average and 90% emergence by 2040. For precipitation the analysis is less conclusive because of high uncertainties in existing reanalysis datasets that also impede an evaluation of the climate models. Model projections suggest hardly any emergence by 2000 but a strong emergence thereafter, reaching 60% by the end of the investigated period (2089). The presented ToE method provides more versatility than traditional parametric approaches and allows for a detailed temporal analysis of climate signal evolutions. An original strategy to select the most realistic model simulations based on the available observational data significantly reduces the uncertainties resulting from the spread in the 65 climate models used. The method comes as a toolbox available at https://github.com/pohleric/toe_tools.

## 1 Introduction

High latitudes experienced pronounced climate change, for example, in the form of warming air temperatures and precipitation regime shifts (Cohen et al., 2018). This manifests in far-reaching impacts on the livelihoods of permafrost communities (Crate et al., 2017), hydrological systems (Gautier et al., 2018; Karlsson et al., 2012; Prowse et al., 2010; Vey et al., 2013; Walvoord and Striegl, 2007; Yang et al., 2002), the evolution of permafrost, including changes in landforms (Boike et al., 2016) and feedbacks with the global carbon cycle (Beermann et al., 2017; Hope and Schaefer, 2016; Schuur et al., 2015). The Lena River catchment in eastern Siberia is one of the largest watersheds in Siberia and provides a major contribution to the Arctic Ocean. It is a perfect example of a permafrost landscape that is prone to and highly sensitive to the impacts of climate change. Available air temperature and precipitation records in

this region extend back more than a hundred years and provide a data base to investigate local trends and variability in climate in more detail.

Despite a general warming trend, a strong spatial and temporal variability is apparent over northeastern Eurasia (Desyatkin et al., 2015; Fedorov et al., 2014b; Gorokhov and Fedorov, 2018), and in the high latitudes in general (Mahlstein et al., 2011). A few locations show no apparent trend over the available long-term records (Fedorov et al., 2014b). Gorokhov and Fedorov (2018) focus on the region of Yakutia (Sakha Republic) and find positive temperature and precipitation trends for the region as a whole for the period 1966-2016. However, spatial and temporal variability is apparent in the form of a stronger warming trend in winter compared to summer (0.4 to 1 ºC decade$^{-1}$ compared to 0.1 to 0.4 ºC decade$^{-1}$), and a negative precipitation trend in the northern region (-8 mm decade$^{-1}$) in contrast to increasing positive trends towards the south (~16 mm decade$^{-1}$). In addition, air and ground temperatures co-evolve with strong spatial heterogeneity (Fedorov et al., 2014b; Romanovsky et al., 2010), potentially associated with changes in regional precipitation and snow cover dynamics (Romanovsky et al., 2010, and references therein).

Such changes propel landscape transitions that are not necessarily linear. For instance, the interactions between meteorological forcing and the ground thermal regime in the permafrost-underlain region are complex due to thermal effects, including phase change in the freeze-thaw cycles and insulation effects of snow covers (Grenier et al., under review.; Walvoord and Kurylyk, 2016). The impacted hydrological cycle already shows a systematic shift towards an increase in the intensity and duration of floods, higher frequency of large floods, and disappearing small floods (Gautier et al., 2018). More changes in the hydrological regime can be expected in the future through geomorphological changes, especially the formation of thermokarst lakes (Fedorov et al., 2014a; Ulrich et al., 2017). Most thermokarst lakes are initiated endorheic but might aggregate and connect to the river network with increasing permafrost thaw.

However, the spatiotemporal variability and heterogeneous evolution of different climate variables raise the question about the regional magnitude of climate change, and how much of the observed variability can be attributed to natural climate variability or to human activities. Additionally, it renders an overarching assessment of how permafrost will evolve under climate change and what this means for the climate system as a whole difficult. The individual analysis of the key variables temperature and precipitation constitutes a first step to approach this problem. The identification of how these variables have individually evolved with respect to their natural variability give insights into the complex, and direct and indirect interactions in the Earth system. Ultimately, this is needed for a comprehensive understanding of the system and an assessment of resulting implications under continuing climate change. It further constitutes a prerequisite for planning and execution of possible adaption and mitigation actions that are needed to cope with the environmental and socio-economic impacts in a timely manner.

As a result, considerable effort has been put in the development of methods to investigate and identify when climate departs or emerges from its natural state or variability (time of emergence – ToE). ToE studies cover a wide spectrum of applications, from the most common climate variables like near surface air temperature and precipitation (Giorgi and Bi, 2009; King et al., 2015; Lehner et al., 2017; Mora et al., 2013), to climate extremes (King et al., 2015; Maraun, 2013; Scherer and Diffenbaugh, 2014), to sea level rise (Lyu et al., 2014). There are several methods to calculate ToE (e.g. Sui et al., 2014, and references therein), depending on the available data sources and the specific purpose of the study. Two major aspects are at the frontline of research. The first concerns the methodology and the second one the data base on which to perform the analysis.

## 1.1 ToE approaches

ToE is defined as the timing when a climate signal, such as temperature or precipitation, permanently exceeds its natural variability (e.g. Giorgi and Bi, 2009; Hawkins and Sutton, 2012). Several existing methods rely on separating signal $S$ (climate change) and noise $N$ (natural variability). Such approaches may require a high level of parameterization (Lehner et al., 2017; Sui et al., 2014), for example, to define natural variability, a threshold for the $S/N$ ratio, or to separate signal from noise. Additionally, some meta-

parameters are needed, such as the size of moving windows or the selection of the period that is considered as reference time (e.g. preindustrial conditions). The variability of a variable within a reference period can be addressed by means of standard deviation (e.g. Hawkins and Sutton, 2012; Lehner et al., 2017; Mahlstein et al., 2011), or by the total observed range in values (e.g. Mora et al., 2013). Signals tested for emergence are somehow filtered to eliminate decadal and lower frequency variability, e.g. by means of moving averages (e.g. Lehner et al., 2017), or polynomial fitting (e.g. Hawkins and Sutton, 2012), and are then compared to the derived reference period variability. Other approaches are based on statistical tests that compare, for example, the distributions between a reference and a target period (King et al., 2015; Mahlstein et al., 2011, 2012). Mahlstein et al. (2012) and King et al. (2015), for example, used the Kolmogorov-Smirnov test (KS-test) with a defined significance level to test the statistical similarity between reference and target period distributions. The KS-test is based on a continuous distance metric, i.e. the maximum difference between two cumulative density functions, but it has so far always been used in combination with a significance level.

All existing ToE methods are by definition a test, either on the exceedance of a *S/N* ratio threshold or based on a statistical significance level. As such, they require a parameterization, which can be a drawback in terms of objectivity and transferability. For instance, dealing with a set of different climate variables may lead to different distribution models, where different dataset record lengths affect the behavior of statistical tests and filtering operations. The development of a non-parametric approach is appealing because results are not impeded by the choice of parameters as in the case of parametric approaches.

## 1.2 Data basis for ToE studies

The second major aspect of ToE research concerns the data basis. Observational datasets facilitate ToE studies that focus on already occurred changes. Direct observational data are the most accurate estimates but come with the downside of data gaps and limited spatial coverage. Reanalysis datasets assimilating observational data provide extended spatiotemporal coverage. Their continuous spatiotemporal coverage is an advantage over meteorological station data, but this comes at the cost of some biases with respect to the real observations (Khan et al., 2008; Serreze and Hurst, 2000). Possible ToE methods for these data types rely on a statistical analysis of their signal's evolution over time. In some cases, including the present study, continuous time-series are compulsory which excludes data from meteorological stations with interrupted observations.
Ensembles of climate model simulation (CS) provide estimates ranging from the past to the future and come with specific data structures. These structures are, in some cases, needed to address the effects of internal climate variability (Hawkins and Sutton, 2012; Lehner et al., 2017; Mora et al., 2013), or allow utilization of preindustrial control runs, i.e. a forcing corresponding to preindustrial conditions (e.g. Karoly and Wu, 2005). The difference between model runs with different anthropogenic forcing scenarios and the control runs can provide an estimate for the effect of anthropogenic forcing on the climate (e.g. King et al., 2015; Knutson et al., 2013; Lyu et al., 2014). However, sometimes large CS ensemble spreads (e.g. Knutson et al., 2013; Koven et al., 2013) introduce considerable uncertainties in ToE estimates (Deser et al., 2012; Hawkins et al., 2014; King et al., 2015). In order to reduce the model spread, a pre-selection of CS can be made based on a comparison between CS and observations, e.g. by means of how the variability of certain variables from observations compare to those in the CS in the region of interest (e.g. Mahlstein et al., 2011). Alternatively, weights can be given to individual CS based on how similar a model internal structure is and how well they represent observational data (Knutti et al., 2017). Identifying a robust and objective function for the selection of CS to reduce uncertainty is, however, difficult as it depends on available observational data and means to assess model similarity (e.g. Knutti et al., 2017; Leloup et al., 2008).

### 1.3 Aims

This study presents a novel ToE approach, allowing investigation of the actual evolution of emergence over time. This differs from other methods in the form of tests that provide either the indication of emergence or not. The approach is applied to near surface air temperature ($T$) and precipitation ($P$) in the Lena River catchment, where changes in landscape (Crate et al., 2017) and hydrological behavior (Gautier et al., 2018; Yang et al., 2002) are already apparent, and for the variables' importance in the hydrological cycle and impact on permafrost evolution. The study is designed to utilize available observational data from meteorological stations, reanalysis data, and an ensemble of CS from the Coupled Model Intercomparison Project phase 5 (CMIP5; Taylor et al., 2012). This multi-step, multi-source approach allows for comparison between obtained estimates from the most reliable (in situ) to the most uncertain (CS) data sources.

We test how such an approach can reduce uncertainty of ToE estimates by introducing a non-parametric method based on an adapted Hellinger distance metric (Hellinger, 1909).
The method does not constitute a test, but a continuous metric that describes how far a climate signal, in form of a time-series, has emerged from its natural variability.
This approach is intentionally non-parametric by design in order to ensure transferability to other scientific fields, and to other variables that inherit any kind of value distribution. Because the metric is derived as a continuous signal, it gives insights into how climate signals emerge from natural variability over time. This provides potential added value to the general question of whether a signal has emerged or not based on a single test. Another strength of this approach is that it facilitates an in-depth analysis of how climate change emerges over time, and, in the process, allows for selecting CS that show an emergence consistent with real observations. Consequently, it allows selecting the most realistic CS.

The succeeding sections present the method in detail, followed by the data sources and obtained results. We discuss the obtained results in the light of previous studies, as well as the unavoidable choices of meta-parameters in detail. The latter comprise the selection of a reference period, which is usually pre-industrial conditions like 1881−1910 in Vautard et al. (2014), or 1860-1910 in King et al. (2015) to identify anthropogenic climate change, and the window width to filter out natural and decadal climate variability of the climate signal. Finally, we present our conclusions on how the presented method provides a versatile tool for ToE studies and how it can reduce uncertainty by the incorporation of observational and reanalysis datasets.

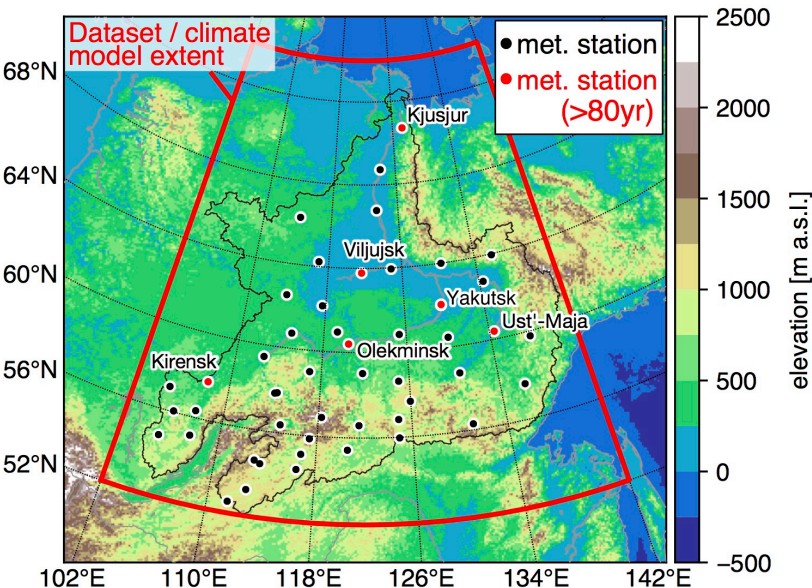

**Figure 1: Lena River catchment (black outline) on topographic map (colour-code) and position of short-, and long-term meteorological stations used to test reanalysis and interpolated datasets. From the long-term stations, Kjusjur has the lowest temporal coverage (less than 10 years) in the reference period 1901-1921.**

## 2 Methods

In the following section we present the methods for ToE detection, sensitivity analysis, and data selection. Our ToE method is a non-parametric metric and thus differs from previous approaches that are parameterized tests for emergence. Our metric describes emergence by measuring how data distributions in continuous target periods have changed with respect to a reference period. Like other approaches, it requires meta-parameter choices, like the start and end point of a reference period and window widths for target periods, for which we will present a sensitivity approach. Finally, the availability of actual long-term observations in the Lena River catchment (Fig. 1) allows validating reanalysis and climate model simulation datasets for their potential to represent the same climate change evolution.

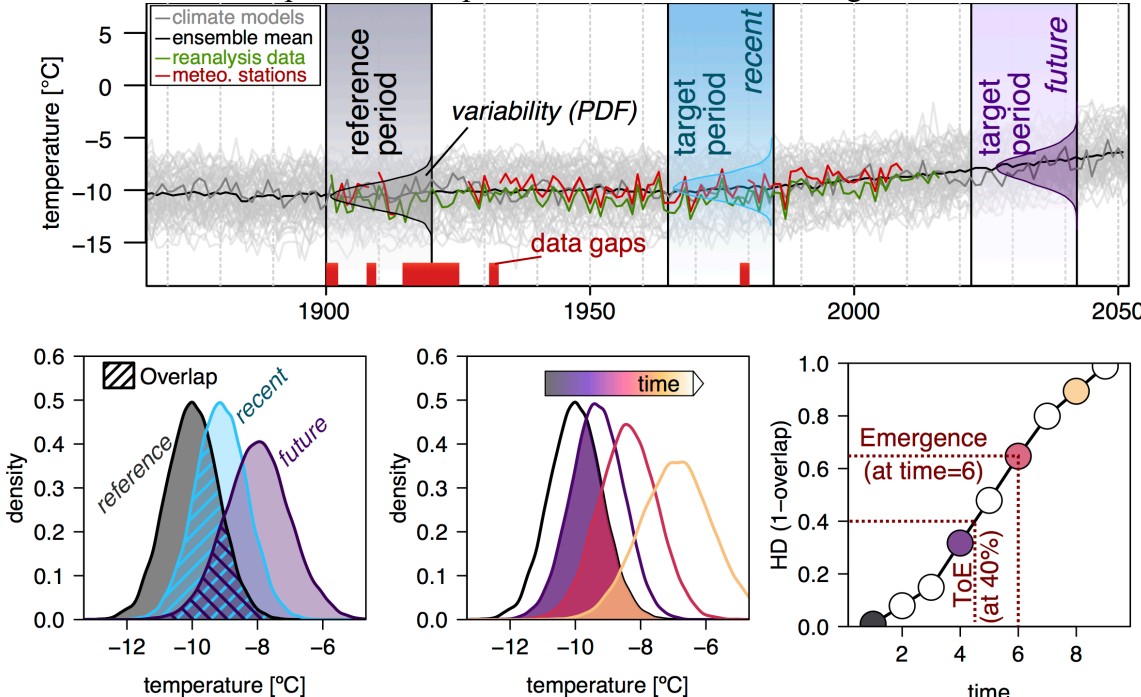

**Figure 2: Schematic of the ToE method based on the Hellinger distance (HD). Top – example of time-series evolution from climate simulations, a meteorological station, and a reanalysis dataset. Natural variability as PDF of the reference period, and two example target periods with a window width of 21 years for recent and future assessment with a sketch of the corresponding PDFs. Bottom – The overlap between the PDFs of the reference and target periods (left), a sketch of PDF evolution over time (middle), and resulting HD as the dissimilarity of the target and subsequent reference PFDs (one minus the overlap). Exemplary determination of ToE for a threshold of 50% emergence, or emergence at a chosen time step, respectively (right).**

### 2.1 Time of Emergence (ToE)

Our ToE method is based on a similarity metric between probability density functions (PDF) described by Hellinger (1909). This metric belongs to a family of distance metrics (Cha, 2007) and can be roughly understood as the geometrical overlap of two PDFs (Fig. 2) (Rust et al., 2010). The method has been used e.g. by Rust et al., (2010) to showcase similarities between distributions of circulation patterns obtained through different climate models.

As we want to describe the dissimilarity, i.e. how far a distribution has emerged from a reference one, we adjust the writing and refer to the metric as Hellinger Distance (HD) according to:

$$HD(Q,R) = \sqrt{1 - \int \left[ Q(x)^{\frac{1}{2}} R(x)^{\frac{1}{2}} \right] dx}, \tag{1}$$

where $Q(x)$ and $R(x)$ are the PDFs of the target ($Q$) and reference ($R$) period, respectively. We use a Gaussian kernel density estimator (KDE) to derive the PDFs from the samples of $Q$ and $R$, and finally calculate the numerical approximation according to:

$$HD(Q,R) = \sqrt{\frac{1}{2} \sum_{i=1}^{d} \left( \sqrt{Q_i} - \sqrt{R_i} \right)^2}, \tag{2}$$

where $Q_i$ and $R_i$ are the densities of the PDFs at position $i$ along a value range that corresponds to the minimum and maximum of the full time series of a variable, extended by the difference between these

extremes in both directions. We use d=200 steps, equally incremented. Tests with more steps and further extended minimum and maximum bounds resulted in insignificant changes (not shown).

The KDE introduces two meta-parameters for the shape of the kernel and the bandwidth (e.g. Scott, 2015). While the kernel shape normally has little impact, the bandwidth can have a strong impact on the obtained
KDE-PDF (Turlach, 1993; Scott, 2015). In contrast, the distance metric in the KS-test does not require a PDF estimate. Therefore, we will show the performance against the KS-test metric, and the sensitivity of our approach with respect to the use of bandwidth selection in Section 4. In our approach we use an automatic bandwidth estimation that is based on Scotts's factor (Scott, 2015), which is only dependent on the number of data points and dimensionality of the data. Therefore, the bandwidth stays fixed in this
work for the calculation of the obtained ToE results as we use fixed window sizes.

HD can take values ranging between 0 (equal distributions/full overlap) and 1 (fully emerged distributions with no overlap). The outline of the method is presented in the schematic (Fig. 2). A climate signal will show a specific data distribution at each time step within a given time window for which a PDF is
calculated (Fig. 2). The HD will increase both if a PDF with a same shape is shifted to higher or lower values, and if its shape changes. The HD is calculated for each time step after the reference period stops. This results in a continuous time-series of HD or level of emergence.  This time-series serves three purposes: 1) A level of emergence can be derived for any given time step, 2) ToE can then be inferred based on a posteriori applied thresholds, and 3) different competing datasets can be tested for consistency
based on their HD evolution (Fig. 2).

We additionally calculate the sign of change because emergence could also occur towards lower values (e.g. less precipitation). The sign (positive or negative) of change is calculated as:

$$sign = \sum_{i=1}^{d}(R_i - Q_i) * bc_i, \tag{3}$$

where $bc_i$ are the actual values at the position $i$ along the extended value range used in (2). We set the
reference period to 1901-1921 and take values for the target periods in moving windows of 21 years too. We test different reference periods and number of years in a sensitivity analysis (see next section). The reference period contains the earliest 21 years commonly available for all datasets. The target periods are taken as a two-sided moving window around each year after the reference period stops, providing a distribution for each time step thereafter. The ToE method is applied independently to the reanalysis data
and each individual CMIP5 CS. We follow previous studies by running our analysis not only at annual scale but also on the seasonal scale (winter – November to March, and summer – May to September) to highlight seasonal differences. Obtained ToE values are given as the year in the middle of the moving window (e.g. a ToE in 2000 corresponds to the target period 1990 to 2010 for a window width of 21 years). We finally test different reference periods and lengths of target periods in a sensitivity analysis
(see next section).

## 2.2 Sensitivity analysis

Our method is non-parametric for the climate change detection but like other methods it requires a set of meta-parameters. These can be divided into two groups. The first group concerns the choice of reference period and the time window for the PDF computation. This is an important issue because climate
variability in the high latitudes is particularly strong (Mahlstein et al., 2011). Thus, it makes sense to test the influence of choosing different reference periods and window widths on the outcome of ToE (Hawkins and Sutton, 2016). We test reference periods ending between 1915 and 1929, and different window widths ranging between 15 and 29 years. While choosing an earlier starting date makes the reference period more 'pre-industrial', it also removes the ability to sample multi-decadal and internal variability. The final
choice is consequently a compromise between the two. Similarly, the choice of longer window widths to choose data distributions is limiting the ability to detect ToE at the end of the time-series. We will present all tested combinations and discuss the derived first-order approximations of uncertainty related to this unavoidable selection of meta-parameters in Sect. 6.2.

The second group concerns the obtained PDFs using a KDE. Two meta-parameters are used for the KDE,
namely the kernel type (e.g. Gaussian, triangular, etc.) and the bandwidth, which determines the

smoothness of the resulting PDF. As mentioned before, the type of kernel has usually a low impact on the resulting PDF, whereas the bandwidth can have a strong impact. We dedicate Section 4 to this analysis by generating synthetic data with exactly controlled intensity changes, and onset of change to test our approach against the distance metric used in the KS-test.

## 2.3 Dataset selection

In order to obtain the most reliable estimates for ToE, the best data choice would be measurements from long-term operating meteorological stations in the Lena River catchment. However, data gaps and a poor spatial coverage demand for alternative data sources to provide a spatially and temporally comprehensive analysis. We thus test three commonly used state-of-the-art reanalysis datasets for their actual representation of in situ temperature and precipitation records. In order to investigate the evolution of climate over the 21$^{st}$ century, we include a collection of CMIP5 climate simulations (Taylor et al., 2012) and test their performance by means of HD evolution (in the past) with respect to the reanalysis data.

The reanalysis datasets are tested against the records from the meteorological stations for near surface air temperature (*T*) and precipitation (*P*) using ordinary least square regression analysis. For each of the 49 stations in the Lena River catchment (Fig. 1), the corresponding pixel-based time-series of either reanalysis dataset is extracted and the performance in terms of explained variance (r$^2$) is evaluated. The best performing dataset is used in the subsequent steps.

For the analysis of ToE in the future, we use both the whole set (n=65) of model simulations but also a subset (n=10). The subset is used to test whether it reduces uncertainty for ToE estimates compared to the use of the entire ensemble. The subset is chosen based on a comparison between HD of reanalysis and climate model simulations. By comparing the HD evolution (0-100%) instead of the actual values, we avoid possible bias issues in temperature and precipitation estimates within the CS. We use the Nash-Sutcliffe efficiency (Nash and Sutcliffe, 1970; Moriasi et al., 2007) as objective function for the selection. In contrast to the r$^2$, NSE adds a penalty for offsets between HD evolutions, according to:

$$NSE = 1 - \left[ \frac{\sum_{i=1}^{n}(Y_i^R - Y_i^{CS})^2}{\sum_{i=1}^{n}(Y_i^R - Y^{Rmean})^2} \right], \tag{4}$$

where $Y_i^R$ is the *i*th HD value of the used reanalysis dataset, $Y_i^{CS}$ is the *i*th HD value of a climate model simulation and $Y^{Rmean}$ is the mean of the HD of the reanalysis dataset (Moriasi et al., 2007).

As we will show in the results, we had to question the validity of reanalysis data in some cases. To ensure confidence in the data we made a further refinement by choosing 5 pixels within the Lena Catchment domain where meteorological stations provide long-term observations and allowed us to verify the quality of the reanalysis. Data records for these 5 stations reach back into the reference period 1901-1921 and cover at least 10 years (see Fig. S1). The corresponding 5 pixels were used to calculate the HD both for the reanalysis and each of the CS. For the sake of completeness, however, we will present the HD evolution of the reanalysis data for the whole study area alongside.

## 3 Data

We focus on the two climate variables *P* and *T* for their importance in the hydrological cycle and for permafrost evolution, and for their relatively good data availability.

### 3.1 Observational data

For observational data we use the All-Russia Research Institute of Hydrometeorological Information - World Data Centre (RIHMI-WDC, http://meteo.ru/) dataset, compiled by Bulygina and Razuvaev, (2012). The dataset comprises 49 stations within the catchment area of the Lena River (Fig. 1). Data were obtained as daily values and averaged and summed to monthly values of *T* and *P*, respectively. The longest records are available for site Yakutsk starting in 1834. All stations within the dataset have record gaps. The dataset provides data only for the locations of the meteorological stations.

## 3.2 Reanalysis data

### 3.2.1 CRUNCEP v7

The CRUNCEP v7 is a global forcing product (ds314.3; Viovy, 2018) used, for example, in the ORCHIDEE-MICT land surface model (Guimberteau et al., 2018). The dataset is derived through a combination of the annually updated CRU TS v3.24 monthly climate dataset (New et al., 2000) and NCEP reanalysis (Kalnay et al., 1996). The CRU TS are based on surface climate data anomalies from different quality-controlled datasets. They are combined with monthly climatologies and interpolated to provide full spatial and temporal coverage. The time coverage is from 1901-2016 in 6-hourly temporal and 0.5º spatial resolution. The data was resampled to monthly averages (sums) of 2m air temperature (precipitation), and to a spatial resolution of 2x2 degrees to match other obtained datasets.

### 3.2.2 Twentieth Century Reanalysis (V2c) (20CR)

The 20CR: Monthly Mean Single Level (Analyses and Forecasts) dataset (ds131.2; Compo et al., 2011) (http:/www.esrl.noaa.gov/psd/data/gridded/data.20thC_ReanV2c.html) contains objectively-analyzed 4-dimensional weather maps and their uncertainty from the mid 19th century to 21st century. The dataset has a temporal coverage from 1851-2011 with a monthly temporal and a 2x2 degree spatial resolution.

### 3.2.3 ERA-20C Reanalysis (ERA20)

ERA-20C is a reanalysis product (ds626.0; ECMWF, 2014) of the European Center for Medium Range Weather Forecast (ECMWF) of the 20th century, from 1900-2011. It assimilates observations of surface pressure and surface marine winds only. A coupled atmosphere land surface and ocean wave model is used to reanalyse the weather, by assimilating surface observations. Data in monthly temporal resolution (monthly means of daily means) in 2x2 degree spatial resolution was obtained.

## 3.3 Climate model data

We use a set of global climate scenarios from the Coupled Model Intercomparison Project phase 5 (CMIP5; Taylor et al., 2012), obtained through the R-package 'esd' (Benestad et al., 2015). The model predictions are biased-corrected through an empirical downscaling approach described in Benestad (2001). All models have historical natural and anthropogenic forcing, and land use for the period 1861-2005, and the concentration pathway 8.5 (RCP8.5) thereafter until 2100. An overview of these model simulations is given in Table S1.

## 4 Performance of HD-based ToE

### 4.1 Comparison of HD to the Kolmogorov-Smirnov distance metric

The most striking difference between the HD-based ToE approach and previous ones is the continuous character of the obtained metric. However, the KS-test also utilizes a continuous metric, namely the maximum distance between the cumulative density functions (CDF) of two samples, in the following referred to as KS-metric. In order to evaluate the additional value of the HD-based approach, we showcase how these two distance metrics compare to each other in terms of sensitivity and accuracy (Fig. 3, Fig. 4). For this, we generated synthetic data with a controlled onset and strength in signal changes. We first use two datasets, one closely resembling a temperature time-series of the utilized climate model datasets (type 1), and one that serves to showcase detection sensitivity (type 2). The type 1 and type 2 data are normally distributed data with a fixed mean and standard deviation (SD) until the breakpoint year (1960). Thereafter, type 1 data have a fixed linear change (slope derived from an arbitrarily chosen pixel of one of the climate model simulations), while the SD stays constant. The type 2 data have a constant mean value after the breakpoint year, but a continuous increase in standard deviation, reaching two times the

reference SD at the end of the time-series. We generate 5000 time-series of each type of dataset and calculate the HD and KS-metric.

Figure 3 showcases one representation of each of the synthetic time-series (upper panels). The distance plots (lower panels) show the median (bold line), inner quartile range (shading), and the 5%-95% percentiles (points) to give a representative assessment of how the two distance metrics perform.

Generally speaking, the HD has a crucial advantage over the KS-metric in terms of continuous change description and also in terms of accuracy. The left panel in Figure 3 shows the co-evolution of HD and KS-metric for a time series with a pronounced trend. A step-function like evolution is visible for the KS-metric. This becomes even clearer if the change in the original signal is smaller (Figure 3 – right). The inner quartile range (IQR) of the HD based on 5000 samples of the time-series is mostly lower than for the KS-metric. Also, the 90% range (5%-95% percentile) reaches overall lower values compared to the KS-metric, as well as less variability along the time axis (the KS-metric changes from a low to a high range within a few years).

The right panels in Figure 3 show a signal with slight changes (gradual increase of the standard deviation) and corresponding distances. The KS-metric is not able to detect the change in a continuous way and only indicates change once a certain threshold is passed. The accuracy, i.e. the range in distance estimates based on the 5000 samples, is very similar in this case. The step-function-like evolution in KS is depending on the sample size, which determines the minimum dissimilarity increase ($1/n$, with $n$ being the sample size). This step-function like evolution is also clearly visible in the example with the strong onset of a trend (Fig. 3 - left). Not shown are the minimum and maximum values that are possible. For KS, these have a wider range because even a very slight shift of an otherwise equal distribution can cause a high KS-metric. However, this does not happen often (not captured by the 90% range).

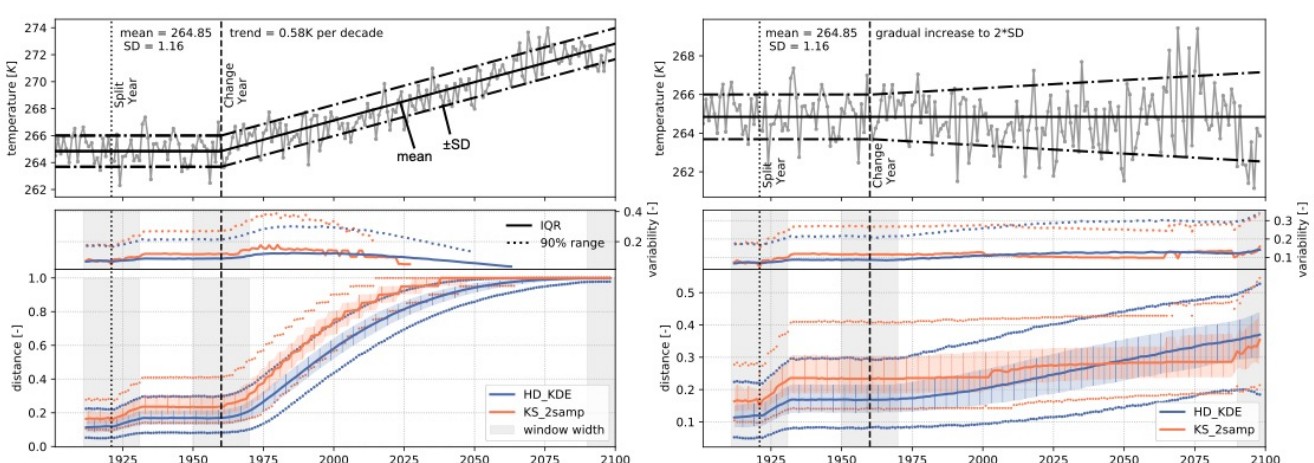

Figure 3: Comparison between Hellinger distance using the KDE (HD_KDE) and the Kolmogorov-Smirnov metric for the two-sample test (KS_2samp). Two synthetic time-series examples (top) and corresponding distance evolution (bottom). Inner quartile range (IQR) and 5%-95% percentile range (middle). Note the different scales in distances between the two types of data.

## 4.2 KDE bandwidth sensitivity

To obtain the reference and target period PDFs, the utilized KDE is using Scott's factor (Scott, 2015) for the automatic bandwidth selection (Fig.4 – left). Even though it stays constant in our analysis with fixed window width and dimensionality, we test the possible impact on obtained HD estimates. For assumed common sample sizes for monthly to annual data between 10 and 100, Scott's factor provides bandwidths between 0.4 and 0.6 (Fig. 4 – left). The resulting change in HD is shown in Figure 4 (middle and right-hand side). We use again the two synthetic time-series from before to show the change in HD. For the relatively large range in sample sizes and resulting change in bandwidths, the overall change is in the range of only 5% for both type 1, and type 2 data. A further analysis of how the bandwidth affects different types of signals is out of scope for this work. We do not explore the effects of different kernel shapes in addition to the bandwidth because of an inferior importance compared to the bandwidth (Bianchi, 1995). However, we also tested the impact of strictly positive and strongly skewed distributions on the approach (Supplementary – Fig. S2). For small differences between such distributions, there is a positive bias

resulting in a HD of at least 0.2 to 0.3. Once the HD reaches 0.3 and above, the bias to the actual HD of distributions becomes less than 10% and becomes independent of the bandwidth.

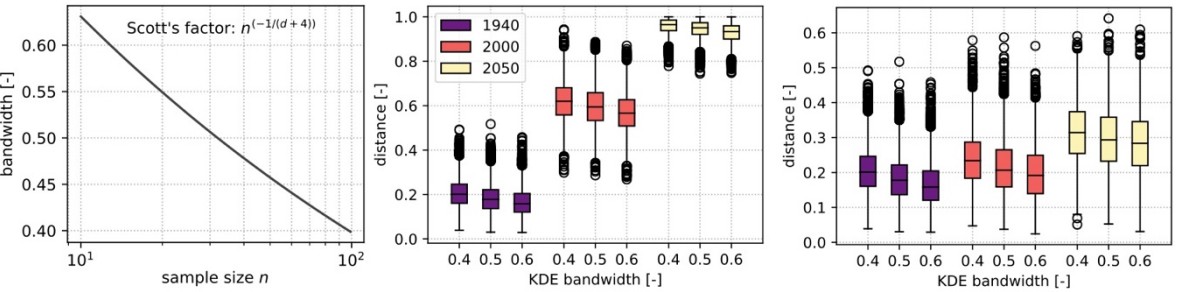

**Figure 4: HD sensitivity to KDE-bandwidth. Automatic bandwidth selection in python's scipy.stats KDE (Virtanen et al., 2020) is based on Scott's factor (Scott, 2015), where n is the sample size and d is the dimension. For 1-dimensional data, sample sizes between 10 and 100 correspond to bandwidths of 0.65 to 0.4. Middle panel is sensitivity in HD for example type 1 (Figure 3 – left-hand side). Right panel is sensitivity in HD for example type 2 (Figure 3 right-hand side) at years 1940, 2000, and 2050.**

## 5 Results

### 5.1 Dataset selection

The comparison of in situ data with CRUNCEP, 20CR, and ERA20 data shows differences in the reanalysis datasets' performances (Fig. 5). $T$ estimates of either dataset explain more than 95% of the variance, but only CRUNCEP's $P$ estimates show high correlation ($r^2 = 0.85$) and limited bias with the observational data. Apart from the poor representation of the other datasets for $P$, 20CR also shows a
15 systematic $T$ under(over)-estimation in spring/autumn (summer/winter) (Fig. 5). CRUNCEP provides the best estimates from the tested datasets for both target variables and is used in the following. As CRUNCEP results partially from direct station measurements, the best match is not surprising, even though we did not test which stations and which periods of the station data are incorporated or rejected. However, some artificial precipitation signals are apparent in the CRUNCEP dataset. These occur mainly
in the northwestern part, where no stations with data records in the reference period exist (Fig. 1, Fig. S1). For this region, the CRUNCEP $P$ data shows a strong artificial, annual repetitive pattern, with probable recycling of the same year, resulting in a very low inter-annual variability (Fig. S3). Here, HD rapidly emerges to more than 40% (Fig. 6, Fig. 7, Video2). While the CRUNCEP $T$ signals do not show a similar pattern that would be easy to identify, the inter-annual variability is also lower in the northeastern
part compared to the rest of the study area (Fig. S3). Whether this implies an area-extensive bias in the CRUNCEP dataset for $T$ is difficult to assess. The resulting differences in the HD for CRUNCEP based on the full dataset vs. the reduced dataset (pixels with validated long-term observations) are displayed side by side in Fig. 6 and Fig. 7 as solid and dashed green lines, respectively; the identified 10 best performing model simulations based on either dataset are shown in Fig. S4 and Fig. S5, and the obtained
NSE statistics derived from this analysis are shown in Table 1. The resulting HD differences are less than 10% emergence for $T$ but in some cases more than 20% for $P$ (Fig. 6, Fig. 7).

The obtained NSE are presented in Table 1 (the corresponding graphs for HD evolution are available in Fig. S6 and Fig. S7). The NSE for $T$ attests a very good representation of the HD for some of the climate model simulations (0.73 to 0.81 for annual values), contrasting with a rather poor representation for $P$
(below 0) (Table 1). Based on this finding, we derive the set of best models based on temperature alone and use the same set for the ToE analysis of precipitation.

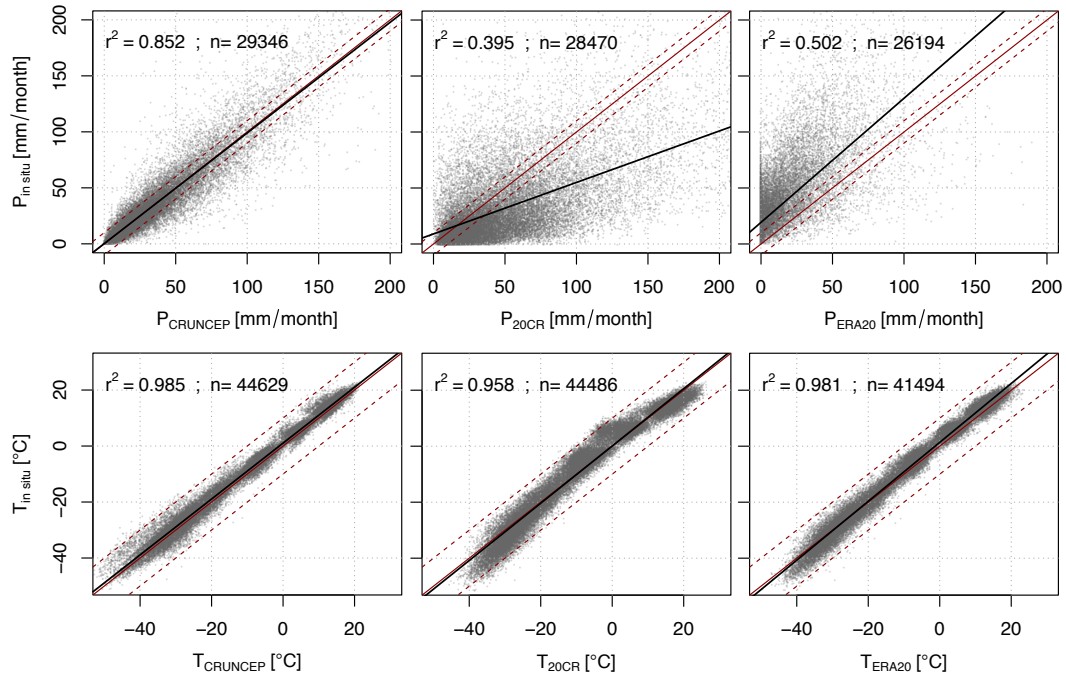

**Figure 5: Comparison of three reanalysis datasets with in situ records (RIHMI-WDC) for monthly values. Red solid line is the 1:1-line; red dashed line is ±10mm for precipitation and ±10 ºC for temperatures.**

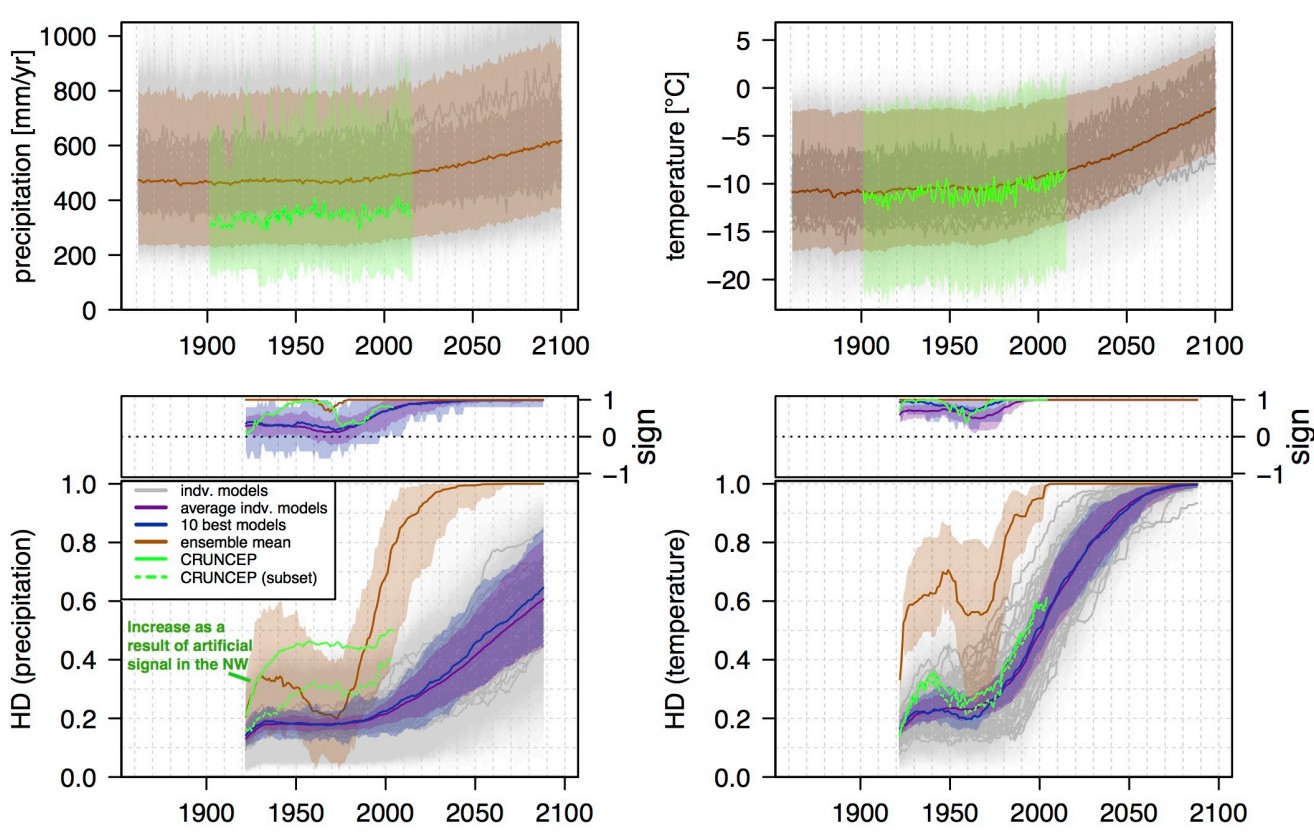

**Figure 6: Area-averaged *T* and *P* signal evolution, emergence as Hellinger Distance (HD), and the sign of emergence. Top - Evolution of summed annual precipitation (left) and mean annual temperature (right) over the entire catchment (red outline in Fig.1). Bottom – Evolution of HD with sign of emergence. Shading indicates the value range over all pixels in the study area. Dashed line for CRUNCEP shows HD evolution based only on the 5 pixels where meteorological stations cover more than 10 years in the reference period to eliminate data issues – see also text and Supplementary for data issues of CRUNCEP. The smoothed signal of the ensemble mean (top) results in a strong and early emergence (bottom) that is not seen in any of the individual models.**

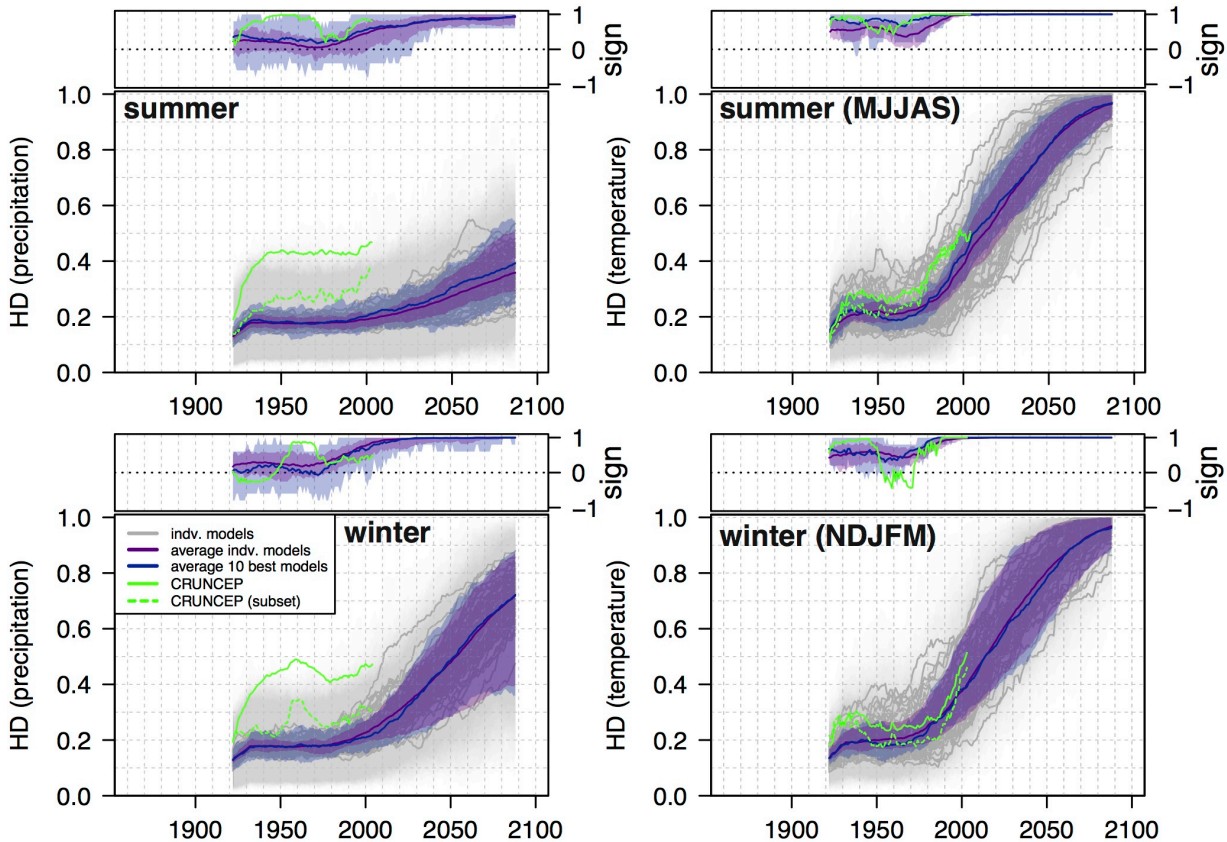

**Figure 7: Summer (top) and winter (bottom) emergence as Hellinger Distance (HD), and the sign of emergence. Shading indicates the value range over all pixels in the study area. Dashed line for CRUNCEP shows HD evolution based only on the 5 pixels where meteorological stations cover more than 10 years in the reference period to eliminate data issues.**

## 5.2 Temporal evolution of temperature and precipitation emergence

The evolutions of area-averaged annual $T$ and $P$ by means of CRUNCEP and the 65 CMIP5 CS, as well as the model ensemble mean are shown in Fig. 6. The CMIP5 ensemble mean temperature is in close agreement with CRUNCEP at annual scale. The ensemble mean for precipitation overestimates the CRUNCEP signal, but some individual CS are close to the CRUNCEP $P$ estimates.

To highlight the effect of our sub-selection method for CS, we present the study area-averaged HD for the different data sources 1) CRUNCEP, 2) individual CS, 3) average of the HD of all individual CS, 4) ensemble mean, and 5) average of the HD of the 10 best CS (Fig. 6). Video1 and Video2 show the spatiotemporal evolution for each of the datasets and seasons, respectively. In particular the HD of the ensemble mean is progressing very differently compared to the other datasets and shows decades earlier emergence (Fig. 6; see Sect. 6.1 for discussion). In contrast, the HD differences for both $T$ and $P$ between the average of all individual CS and the average of the 10 best CS are the lowest and show a similar evolution. Individual CS may show a very different evolution and different regional patterns, which is also highlighted in Video1 and Video2. The videos include the single best performing CS to showcase the higher spatiotemporal variability of individual CS compared to the averaged ones.

The $T$ signals show the most prominent evolution and the most significant emergence. The emergence patterns for CRUNCEP and all individual CS are very similar (Fig. 6). The HD shows a continuous increase starting in the 1960s. For CRUNCEP this increase is preceded by an initial HD increase at the beginning of the target period and stagnation thereafter until the 1960s. In contrast, HD increase based on individual CS indicates little change (<30%) until the 1960s and 70s with respect to the reference period.

The CRUNCEP signal emerges above 60% by 2004 (last data point). The average HD of individual CS and the 10 best CS reach 90% emergence in the 2040s, and near 100% emergence by the end of the time series (2089). In stark contrast to that is the HD based on the ensemble mean, which shows a 100% emergence already by 2004.

For *P*, the evolution of the CRUNCEP signal and individual CS, as well as the corresponding HD show more significant uncertainties and are less well-defined (Fig. 6). The ensemble mean shows an emerging positive signal from 2000 onwards. The HD for CRUNCEP shows early strong emergence in the northeastern parts and to a lesser degree regionally across the entire domain (Video2), which is related to the before-mentioned data issues in the CRUNCEP dataset. The average HD of all individual CS, and of the 10 best CS show an almost identical evolution until the 2000s when the HD shows a distinct departure reaching around 60% emergence by 2089.

The sign change for both *T* and *P* is permanently positive once 40% and 30% emergence is reached, respectively. Before that, until the 1970s, around 60% to 80% of the pixels show a positive trend for *T*, and 50% to 60% for *P* (Fig. 7).

The seasonal (summer and winter) evolutions show generally the same trend as the annual ones but some differences are apparent. Most striking is the stronger regional variability in HD for *T* in winter compared to summer (vertical shading in Fig. 7). For *P*, the seasonal difference is striking. An overall emergence of ~70% in winter compares to <40% in summer. The corresponding area-wide mean ToE and corresponding changes in *T* and *P* are summarized in Table 2. The biggest ToE differences between summer and winter are apparent for *P* (20-29 years), whereas for *T* there is only a maximum difference of 1 year. ToE of *T* for annual values is 11-15 years earlier compared to summer and winter.

For *P*, the annual ToE is in between the winter (earliest) and summer ToE.

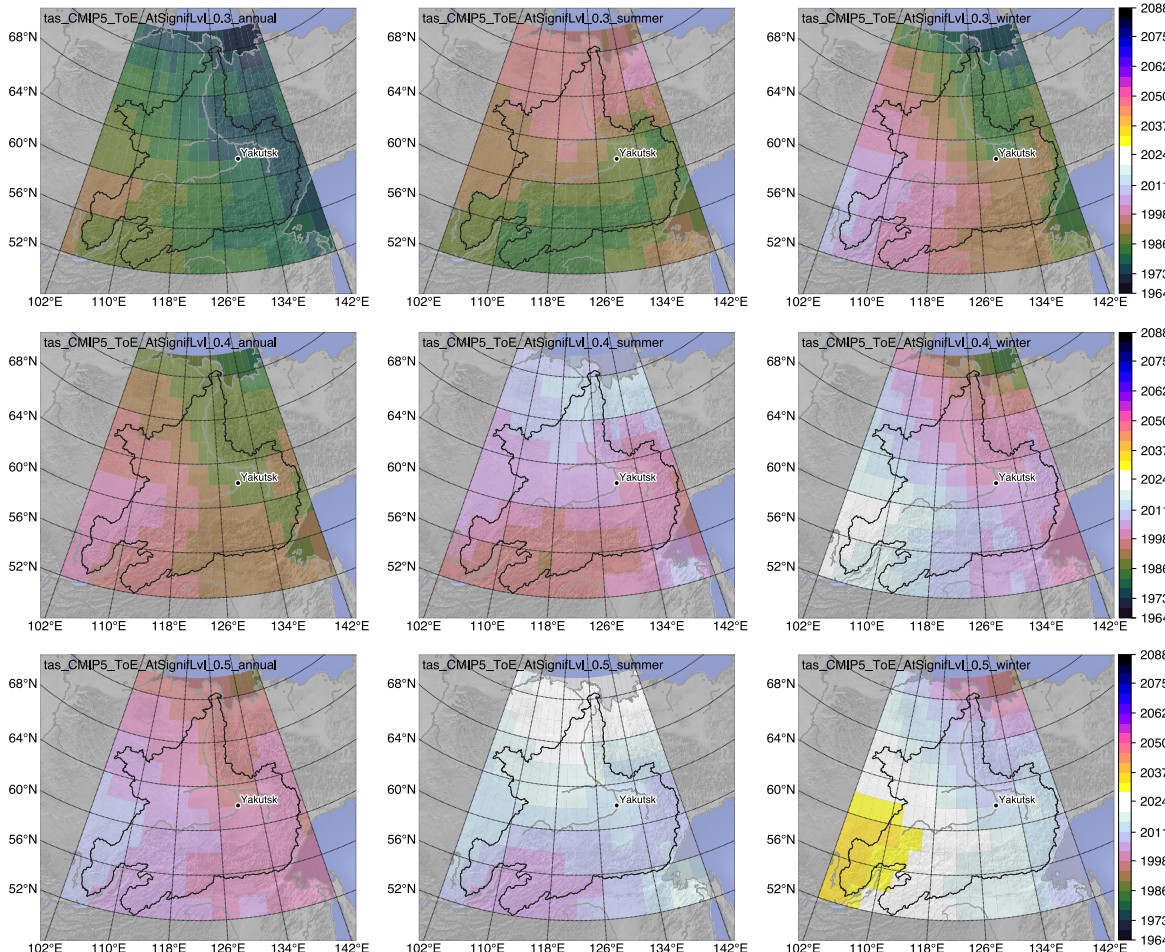

**Figure 8: Time of emergence for temperature according to 30% (top), 40%(middle), and 50%(bottom) emergence for annual (left), summer (middle), and winter (right) values. Values are the mean over all individually determined ToE for each of the 65 climate simulations.**

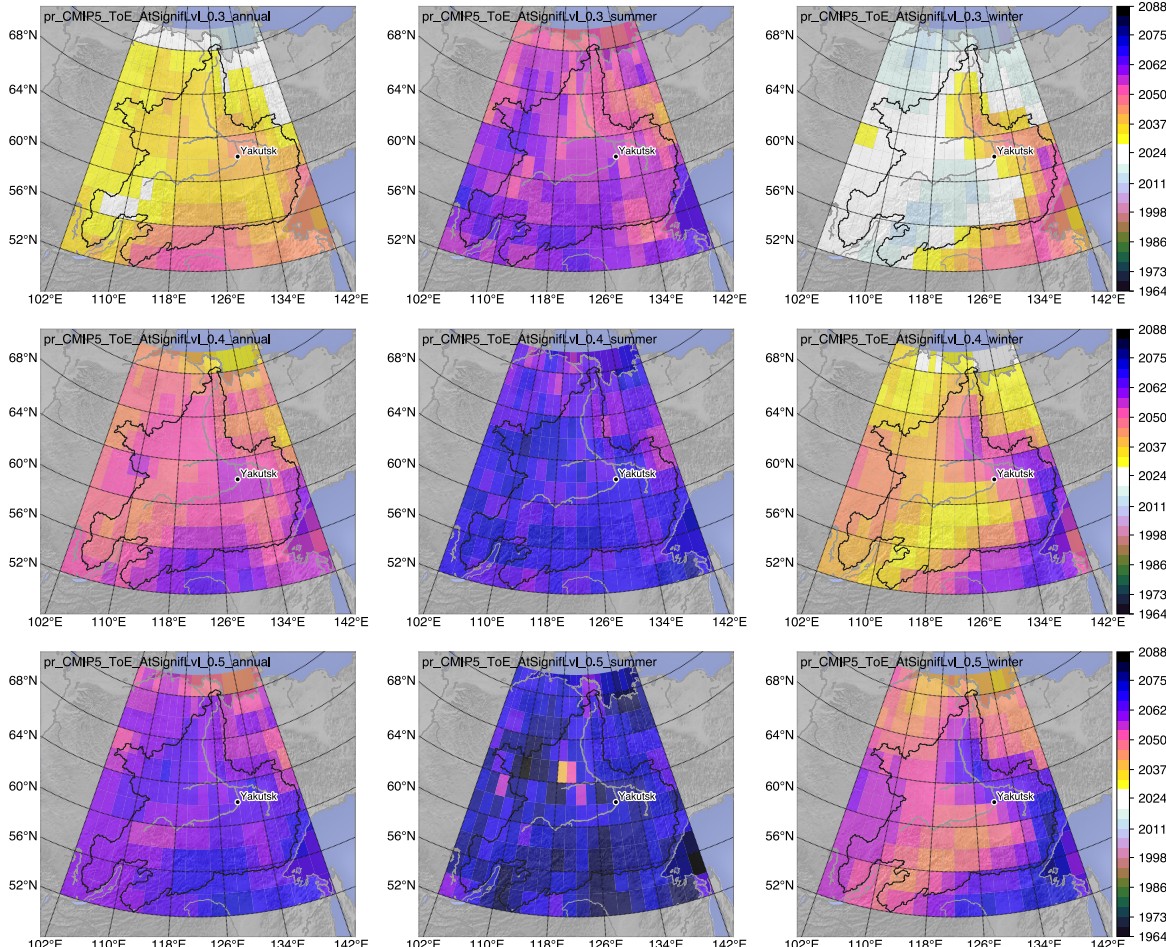

**Figure 9: Time of emergence for precipitation according to 30% (top), 40%(middle), and 50%(bottom) emergence for annual (left), summer (middle), and winter (right) values. Values are the mean over all individually determined ToE for each of the 65 climate simulations. Artefacts at 50% emergence in summer (earlier ToE than for 40%) are due to limited number of model simulations with emergence.**

## 5.3 Spatial and seasonal variability

The spatial variability in ToE over the study area (vertical shading in Fig. 6, and Fig. 7) is displayed as maps in Fig. 8 and Fig. 9 for three different emergence levels (30-50%) and the three temporally aggregated periods (annual, summer, winter). The corresponding changes in $T$ and $P$ for a ToE at a given emergence level are shown in Fig. 10 and Fig. 11. Due to the nearly identical evolution of ToE based on the mean HD of either all individual CS, or the 10 best CS (cf. Fig. 6, Fig. 7, Video1, Video2), we only display the results for the former.

The annual and winter analyses for $T$ show generally earlier ToE in the northeast compared to the southwest (Fig. 8). The summer pattern is almost reversed with earlier ToE in the south and later ToE in the north.

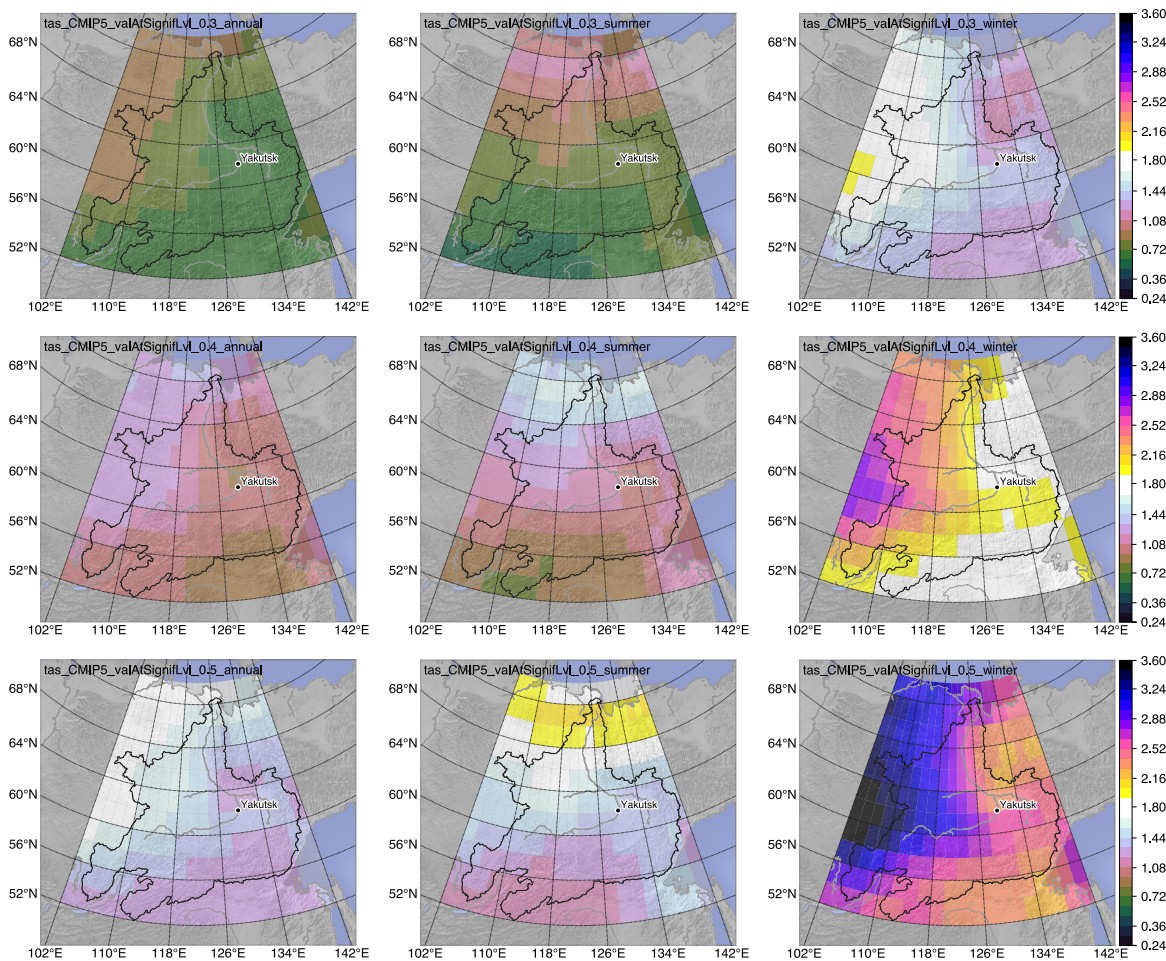

**Figure 10: Temperature change (ºC) corresponding to 30% (top), 40%(middle), and 50%(bottom) emergence for annual (left), summer (middle), and winter (right) values. Values are the mean over all individually determined changes for each of the 65 climate simulations.**

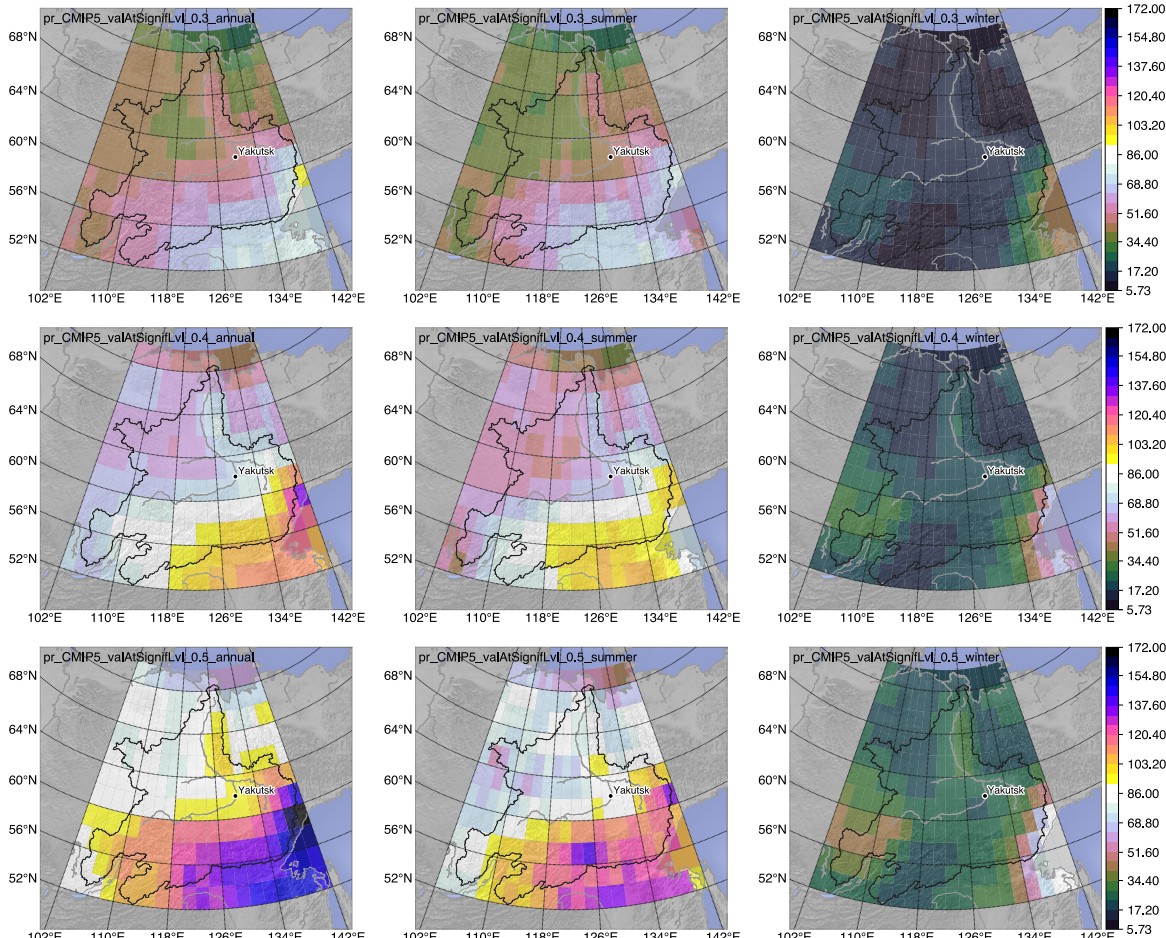

**Figure 11: Precipitation change (mm yr⁻¹) corresponding to 30% (top), 40%(middle), and 50%(bottom) emergence for annual (left), summer (middle), and winter (right) values. Values are the mean over all individually determined changes for each of the 65 climate simulations.**

The strongest variability in ToE for a given emergence level can be seen in the winter analysis, where the earliest and latest ToE can lie more than 30 years apart. Corresponding changes in $T$ for a given ToE strongly depend on the spatial location and the season. For example, 50% emergence in the $T$ signal corresponds to 1.2 ºC for the annual analysis in the south, contrasting with 3.6 ºC for winter in the west (Fig. 10). Based on the temporal aggregation, an up to two-fold difference in $T$ change can be observed for a given emergence level (annual vs. winter) (Fig. 10, Table 2).

For $P$, there is also a N-S gradient towards later ToE observable (Fig. 9). In addition to that, a pronounced later ToE along the eastern catchment boundary is visible in winter, whereas annual and summer ToE do not show such a pronounced feature. Most striking for $P$ are the strong ToE differences between the seasons, with locally up to 50 years earlier ToE in winter compared to summer.

Corresponding changes in $P$ for a given ToE relate to the pronounced seasonality with the highest moisture supply in summer. This results in an up to four-fold stronger increase in summer compared to winter, adding up to the roughly two-fold regional differences (Fig. 11).

Comparison of the ToE between the two variables $T$ and $P$ shows strong differences that locally reach 80 years (Fig. 8, Fig. 9). The area-averaged differences between 30% and 50% emergence correspond roughly to a doubling of change in $T$ and $P$ for any season (Table 2).

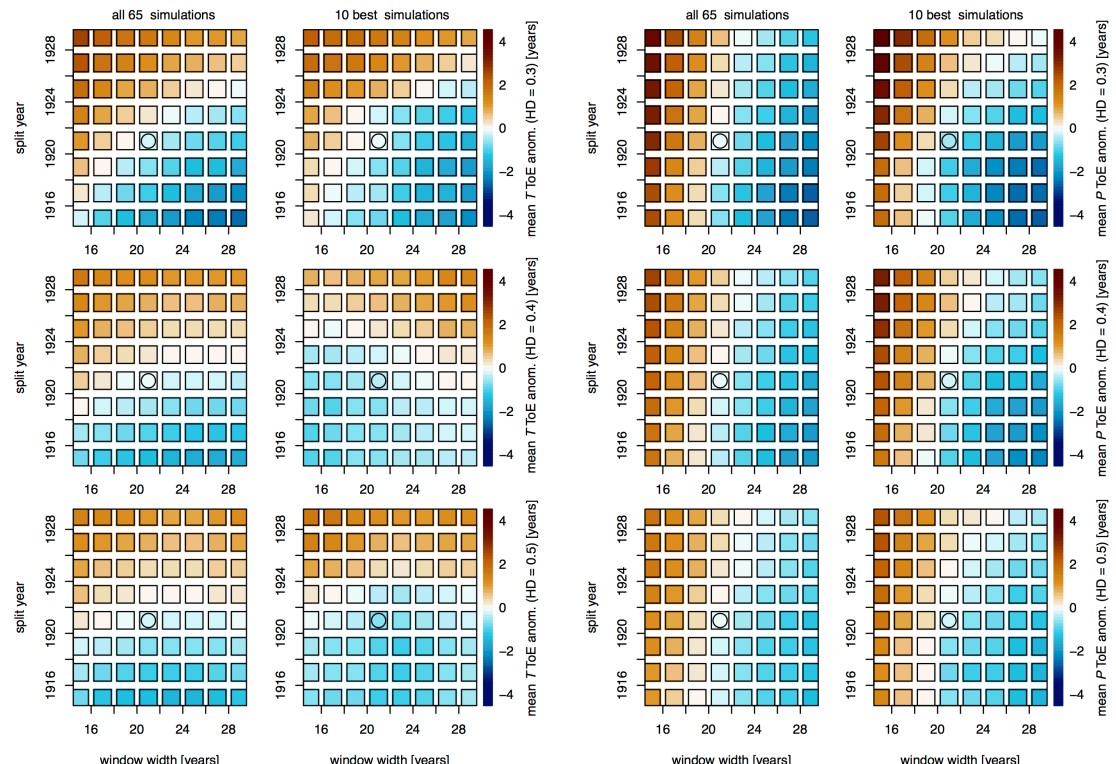

**Figure 12: Impact of window width and split year on ToE for *T* (left two columns) and *P* (right two columns) as mean deviation from the mean over all combinations of window width and split year. Individual left columns for all 65 CS and individual right columns for the subset of 10 best CS. Rows – different emergence levels (30-50%). The average ToE and standard deviations are available in Fig. S8 and Fig. S9.**

## 5.3 Sensitivity Analysis

Although the developed approach for ToE computation relies on PDFs and is basically non-parametric, the method requires two meta-parameters with a potential outcome on obtained emergence and ToE: target period (split year) and window width to calculate the PDFs of the reference and target periods (Fig. 12). The model simulations' internal maximum deviations for the tested meta-parameter combinations of window width and the end of the target period (split year) are around ±4 years on average, for both for *T* and *P*. In contrast, the inter-model differences are up to 70 years at low emergence levels, which can be seen in Fig. 6, and Fig. 7. No particularly abrupt increase in ToE for a specific year or window width is apparent. The more dominant parameter on the outcome of ToE is the window width for *P* as can be seen in the horizontal gradient in Fig. 12. For *T*, a stronger variability between the simulations and at different emergence levels is present (not shown) and the resulting average sensitivity in Fig. 12 is less pronounced than for *P*. A slight change from a gradient with later ToE for a late split year and short window width at 30% emergence to a generally later ToE mainly based on split year length can be seen. The latter is represented by the vertical gradient. No particular year or window width can be identified to have a significant impact on the ToE estimates for either variable.

For both variables, the sensitivity to either meta-parameter based on all 65 CS or the 10 best CS is equally low. However, the standard deviation of ToE estimates is reduced by up to 6 years for the case of the 10 best simulations (Fig. S8, Fig. S9). Derived ToE sensitivities for the full set of CS, and the subset are very similar and reflect the similarity presented in Fig. 6 and Fig. 7 (cf. Fig. S8, Fig. S9). The average patterns for both the 65 and 10 best CS also largely resemble the pattern for CRUNCEP (Fig. S10). However, a sharper contrast for CRUNCEP between split years and window widths, and a stronger impact on the range in ToE are apparent. ToE estimates for low emergence levels reach up to ±9 years for *T*, which is also the maximum range found amongst all individual CS for different emergence levels. In summary, the found maximum variability resulting from the meta-parameter choice is very low (±4 years) in comparison to the inter-model variability (up to 70 years), and is well below commonly reported ToE bin sizes, i.e. time intervals (~20 years) to classify a regions' ToE.

## 6 Discussion

The results showcase a strong variability between the temporal evolution of emergence and derived ToE of the two tested climate variables $T$ and $P$. Large differences also occur between the three temporal aggregations: annual, winter, and summer. These differences highlight the complexity in the climate system and emphasize that there cannot be a single answer to the general questions if, and how much climate change has emerged in eastern Siberia.

### 6.1 Method

The ToE method applied in this study provides an innovative way to investigate climate change evolution and its emergence. Different from existing ToE methods that rely on tests, based either on exceedance of a $S/N$ threshold (e.g. Hawkins and Sutton, 2012) or a statistical significance level (e.g. Mahlstein et al., 2011). The new approach provides a continuous measure of emergence. This has advantages and disadvantages to previous methods. Striking benefits are that it facilitates comparison of the evolution for different datasets (Fig. 6, Fig. 7), allowing to rank and select climate simulations whose emergence signatures correspond the closest to observational data. This is a big difference with respect to pre-selection procedures based on statistical comparison (e.g. Mahlstein et al., 2011), or on weighting schemes that compare model similarities and the ability of CS to represent observational data (Knutti et al., 2017). The expected downside of the developed method is the need to define an emergence level a posteriori in the present case. However, it should be kept in mind that other methods also require a threshold in form of a $S/N$ ratio, or as a significance level for statistical tests. In our case, the information about the significance is directly provided by the value of emergence and allows answering questions like what a halfway-emerged climate looks like compared to initial conditions. The main difficulty might lie in finding a connection between the level of emergence of a climate variable under investigation and how it relates to possible environmental and socio-economic impacts. This certainly requires expert knowledge of already occurred or observed on-going changes that involve complex interactions in permafrost landscapes.

The Hellinger distance shows particular advantages over the KS-metric in direct comparison. It is able to detect very small changes (Fig. 4) and the detection limit is not dependent on the sample size as is the KS-metric, which produces a step-function like evolution. However, the approach also comes at the price of a higher computational cost, e.g. through the calculation of PDFs using a KDE.

A very surprising finding is that independent of whether we calculate the average HD of the subset of best models or of all CS, the derived emergence and ToE estimates show only a few years difference, despite several decades differences between HD of individual CS (Fig. 6, Fig. 7). The strongest impact of the sub-selection on the results is the reduced ToE variability between individual HD evolutions (Fig. S8, Fig. S9). A rather low impact on ToE from choosing a preselected number of CS that best match the observations was also reported by Mahlstein et al. (2011). In stark contrast, applying the method on the ensemble mean yields significantly earlier and stronger emergence (Fig. 6), resulting from the extreme narrow range of the filtered signal. This is similar to the muting of internal climate variability through having multiple model runs using the same climate model (e.g. Deser et al., 2016). Resulting PDFs are very narrow and exceedance occurs more rapidly than we can observe in any of the individual signals, including the CRUNCEP (Fig. 6, Fig. 7). In the present case, this muting is inconclusive because it results from the averaging over different climate models with different internal variability. As our method is designed to specifically detect the change of a signal with respect to its natural variability, the presence of variability is a prerequisite.

The development of our method was made with the intention to have a wide range of applications, including nearly all types of time-series data. Like the application of the KS-test (King et al., 2015; Mahlstein et al., 2012), PDFs can be obtained for any data distribution and their overlap as a measure for emergence can be easily understood. The resulting emergence as a time-series provides the advantage over previous methods to investigate how a signal has emerged in detail. How different datasets and the

utilization of different temporal resolutions (e.g. monthly data) affect the determination of ToE should be explored in more detail in the future.

## 6.2 Sensitivity

The expected uncertainty from the needed meta-parameter selection of window widths and reference period (cf. Hawkins and Sutton, 2016) has a rather negligible impact on the overall outcome of ToE compared to the differences resulting from the spread between individual CS (Fig. 12): the study-area averaged variability of ±4 years across all meta-parameter combinations contrasts with up to 70 year-differences between individual CS. The analysis revealed some systematic patterns in the form of a dominant gradient in vertical direction for $T$, and a horizontal gradient for $P$ (Fig. 12), providing insights
into some important aspects related to the data and the method itself.

A longer reference period and accompanied later ToE, as can be seen for temperatures (Fig. 12), indicates mainly that a trend towards increased values at the end of the reference period is present. Extending the reference period provides a wider PDF and higher values. The target periods will stay longer overlapping and ToE occurs later. A reverse situation with lower values towards the end is apparent in single cases
only (not shown) so that the vertical gradient is reversed.

The gradient towards later ToE for smaller window widths for precipitation (Fig. 12) is somewhat counter-intuitive as a small window size implies more variability. It results from local minima (low precipitation years) that strongly impact the PDFs in the target period. They can thus again become similar to the PDF of the reference period. Consequently, an earlier continuous exceedance is not treated as
permanent and the finally obtained ToE is later for a small window width. Longer window widths will cause the extreme values to have a less significant impact on the PDF. The resulting dissimilarity stays above the threshold and the derived ToE is earlier even if the initial threshold was crossed later. The same reason seems to cause the earlier emergence for annual $T$ and $P$ values, where a single extreme month has a relatively low impact on the annual PDF compared to its stronger impact on the seasonal PDF (Table
2).

We assume a low impact on the uncertainty from the KDE-based determination of the PDFs for the HD calculation. Even though we have not tested the impact of the bandwidth on the different window widths in the sensitivity analysis, we have shown that for similar data and even larger ranges in sample sizes (10 to 100), the overall uncertainty is in the range of ±5% (Section 4, Fig. 4). Since our approach also shows
a low bias for non-normal distributions once a certain HD is reached (in the presented case 0.3; Section 4), we assume a wide range of applications. An option to turn off the automatic bandwidth determination is possible in the code implementation. This provides the possibility to test how this meta-parameter affects other types of data.

In summary, the sensitivity analysis is a valuable and relatively easy to apply tool to explore how a
35 specific dataset and a combination of meta-parameters influence ToE estimates.

## 6.3 Data

Our initial selection of reanalysis data through comparison with observational data has shown good agreements for $T$, but except for CRUNCEP a very weak representation for $P$. In combination with the systematic bias for warmest and coldest temperatures for 20CR (Fig. 5), this also requires a cautious
selection of CS based on observational data in the region. As CRUNCEP results from interpolated observational data, the good match is not surprising. How well this dataset represents the actual conditions for the times where there are no measurements available remains unsolved. The apparent recycling of a single year in the CRUNCEP time-series (Section 5.1) and the resulting standard deviation close to zero (Fig. S3) indicate that the data is biased or unreliable in the north-western part.
Interestingly, the selected best CS are from different models within the ensemble. That is despite some of the selected best CS belonging to models that are represented with several runs in the ensemble (cf. Fig. S5, Fig. S4, Table S1), meaning that internal climate variability within the models of the ensemble plays

an important role for the case presented here, and potentially other ToE methods. It also stresses the benefit of ensembles to include multiple runs of a model, because it additionally helps other approaches to identify internal climate variability (Deser et al., 2016). While the HD comparison to select CS for *T* shows very good matches (Table 1, Fig. S6), the imperfect matches for *P* imply a high level of uncertainty that is difficult to assess (Fig. S7). The best indicator suggesting some reliability is the fact that the sensitivity for CRUNCEP (Fig. S10) shows similar patterns compared to the ensemble of CS (Fig. 12). This pattern match can be interpreted as both datasets having a similar variability and distribution of extreme values, as well as an overall similar trend, as discussed in Sect. 6.2. However, the presented results for *P* should be treated with caution. Climate model simulations and reanalysis data need to be improved to derive regionally reliable estimates, which in turn are needed to investigate the physical processes in the Earth system that can aid decision making.

## 6.4 ToE

ToE values are with respect to the reference period (1901-1921) and thus slightly later than otherwise chosen pre-industrial reference periods (e.g. 1881−1910 in Vautard et al., 2014, or 1860-1910 in King et al., 2015) but longer than in ToE studies focusing on observational data. There is no way to avoid this selection in the current study. The chosen period is the earliest possible one to have a basis for the comparison of the observational data and CMIP5 model simulations.

Data issues are almost always due to the lack of data or data quality (e.g. Hawkins and Sutton, 2016). The sensitivity analysis (Fig. 12) shows that choices of reference periods between 1901-1915 up to 1901-1929 have relatively small impact on the obtained ToE and that uncertainties from the spread in individual CS are an order of magnitude higher. Since we report the emergence as a continuous signal, the question arises when this signal should be considered as significantly different with respect to the reference period. In other words, how strongly does a PDF need to change from its initial shape and position to indicate a significantly emerged climate? An obvious way is to compare obtained results with previous ToE studies and with reported changes in climatic variables.

King et al. (2015) reported ToE for the region of the Lena River between 1980 and 2000 for summer temperatures, and between 1980 and 2000 and in a few occasions between 1960 and 1980 for winter temperatures. These ToE were obtained through a KS-test and using 1860-1910 as a reference period. The reported ToE correspond to the pronounced onset in the HD signal (Fig. 7) and an emergence level of around 30% (Fig. 8). King et al. (2015) further report ToE for winter precipitation between 2000 and 2020 in the lowlands, and 20 to 40 years later in the east and southeast. The same spatial pattern is derived with our method. Again, the timing corresponds to an emergence level of around 30%. Mahlstein et al. (2011) reported temperatures corresponding to the statistically significant identified changes using the KS-test with a reference period of 1900-1929. A direct comparison is difficult as they report these temperatures for countries. However, their identified value of 1.1 ºC for summer temperatures for Russia corresponds to the 30% emerged signal in our study (Fig. 10).

Comparisons with temperature (Desyatkin et al., 2015; Fedorov et al., 2014b) and precipitation trends (Gorokhov and Fedorov, 2018) are somewhat complicated due to different starting points of the datasets. Trends in Gorokhov and Fedorov (2018) are with respect to the 1966-2016 period. As indicated in Fig. 6, the study-area wide precipitation signal shows relatively high values in the 1960s, with a positive emergence for CRUNCEP and a decline thereafter (Fig. 6, Fig. 7). The derived trends in Gorokhov and Fedorov (2018) start in this positive emergence and are consequently depicting a negative trend in the northern regions (~-8 mm decade$^{-1}$), where precipitation changes according to the CS are lowest (Fig. 11). Gorokhov and Fedorov (2018) still find increasing positive trends towards the south (~16 mm decade$^{-1}$). This north-south gradient is reflected by our results (Fig. 10) even though we cannot associate any trend value with a derived emergence level.

Fedorov et al. (2014b) reported generally stronger positive trends for temperatures in the eastern and southern mountain regions in our study area; and lower trends in the lowlands and towards the east. Some general overlay of earlier ToE (Fig. 10) is visible for stronger trends, and vice versa. However, weaker trends in the most northern part and one of the strongest trends for Yakutsk in the lowland render a conclusive comparison difficult. Fedorov et al. (2014b) use a dataset with variable station record length,

which might explain to some degree the discrepancies. In the end, such differences are expected given the variability in the CMIP5 model simulations and individual offsets to the CRUNCEP (Fig. 6).

In relation to such evolutions in $T$ and $P$, ground temperatures and hydrological conditions are especially impacted. Fedorov et al. (2014a) pointed out that in the 1950s high ground temperatures might have initiated thermokarst lake formation. Identification of periods in which a triggering event initiates a state change are not included in any ToE method, despite their potential for landscape changes, that in turn has far-reaching impacts on permafrost evolution (Crate et al., 2017; Grenier et al., 2018; Walvoord and Kurylyk, 2016; Westermann et al., 2017). However, Fedorov et al. (2014a) also mention that despite the early initiation, the main progression of lake formation occurred in the 1990s, which represents the previously-mentioned time period where emergence levels reach 30%.

Warmer summer temperatures of 1 ºC to 2 ºC in the future in summer (Fig. 10) imply a strong impact on the hydrology by means of potential evapotranspiration increase, and the evolution of thermokarst lakes. It is, however, difficult to exactly identify how the co-emergence of $T$ and $P$ at different rates (Fig. 8, Fig. 9) will affect the evolution of thermokarst lakes that are currently in equilibrium between precipitation and ground ice melt water input, and evapotranspiration output. Karlsson et al. (2012) point out that an increase in $T$ would likely increase lake bodies due to the more important input from ground ice melt. This is in agreement with conclusions by Fedorov et al. (2014a) for the formation of new thermokarst lakes. However, old Alas lakes with reduced input from ground ice melt might undergo a reduction if evapotranspiration increases more than total precipitation influx. More recently, Ulrich et al. (2017) have shown through multiple regression analyses that, in particular, increasing winter precipitation and winter temperatures control lake area changes of young and old thermokarst lakes in Central Yakutia. As these two variables show the strongest emergence (Table 2), an increase in thermokarst lake area, and a resulting overall change in the hydrological system, should be expected.

Mean annual discharge of the Lena River has only increased significantly in the most recent 2006-2012 decade (Gautier et al., 2018). However, late spring discharges during the ice break up had already experienced a strong increase a decade earlier (1996-2005). These periods lag the ToE presented for $T$ but precede the ToE for $P$ at 30% emergence (Fig. 7). Taking into account the mutual interactions between temperatures and precipitation, which results in snow cover and ground thermal insulation as well as snow stocks for melt (Grenier et al., in review; Karlsson et al., 2011; Westermann et al., 2017), systematic changes should occur as a result of the two. The onset of winter $P$ emergence in the 1990s and more strongly thereafter would provide a possible explanation. It would also not contradict the strong positive emergence for $P$ in the 1950s and 1960s (Fig. 7) that has not resulted in detectable flood events. The HD and the signal of change for the CRUNCEP data show that more precipitation (positive signal) occurred alongside more negative temperatures (negative signal), which would counteract strong melting events.

The implied changes in $T$ and $P$ at different emergence levels will certainly have significant impact on various environmental and socio-economic aspects. How much these changes, at 50% emergence and more, and at different seasons will impact the complex hydrological system is difficult to assess and should be explored further in the future. Such assessments require, however, a continuation and advancing in the modeling of cryo-hydrological systems that allow for a better understanding of how the climate variables affect the involved processes (Grenier et al., under review.; Walvoord and Kurylyk, 2016). This, in turn, requires for the continuation of measurement efforts in the large, remote, and difficult to access arctic regions, where observational data is sparse.

## 7 Conclusions

We developed a novel method for the determination of climate change emergence. Its non-parametric character allows application on data with different types of data distribution, which we show-cased for $T$ and $P$ in the Lena River catchment, and using synthetic datasets. The strongest biases were found in a synthetic dataset for low changes in PDFs when the distributions are strictly positive and heavily skewed, which might be expected for high-frequency data, like hourly precipitation. Even then, once the distributions show a HD of 0.3, these biases fall below 10% and attest a large application range of the approach. Unlike other ToE methods that rely on a threshold or statistical test, our method provides a

continuous signal of emergence. This facilitates an extended analysis of the progression of climate change signals and provides a useful tool for comparing datasets regarding their similarity in describing climate change. It comes with the need of applying a threshold a posteriori. Comparison with ToE estimates from other studies indicates that an equivalent ToE occurs at an emergence level of around 30% for both $T$ and $P$.

A comparison of three commonly used state-of-the-art reanalysis datasets with observational data from meteorological stations has revealed a generally good agreement for $T$, but only the tested CRUNCEP data provided $P$ estimates with little bias. Even within this dataset, we found artificial behavior in the time period 1901-1921 for the $P$ estimates, probably due to the limited number of meteorological stations operating at that time. In combination with the $P$ intensity bias of many of the CS, conclusions on the emergence of $P$ are rendered uncertain.

Our method allowed us to compare the evolution of emergence of $T$ and $P$ from CRUNCEP with those of 65 climate model simulations taken from a CMIP5 ensemble. This provides an alternative to pre-selection methods based on dataset statistics, or weighting schemes for climate models and simulations.

We obtain surprisingly similar emergence times independent of using either the mean emergence of all simulations or from our sub-selection of the 10 best performing simulations. On the contrary, individual models show estimate differences up to 70 years at low emergence levels. This provides confidence in using large enough ensembles rather than somehow chosen sub-selections to identify ToE if no or insufficient observational data is available. Nonetheless, the selection method presented here might provide means to discriminate the most reliable data sources in other more documented regions or contexts. The conclusion to include full climate ensembles rather than single simulations is supported by a consistent similarity between the full set and the subset of CS in all applied cases ($T$ and $P$ for annual, summer, winter). The differences in derived emergence for reanalysis and climate simulations, however, stress the need for model improvements and an effort for continuous observational data, which can be comprehensively utilized in the presented approach.

Finally, the methodology should be explored in the future to analyze further impacted variables (e.g. ground temperatures and hydrological conditions) in the complex cryo-hydrological system to identify spatiotemporal links. Ultimately, these are needed to derive an understanding of how and when climate change will impact the numerous aspects of this system.

**Code availability**

The main code to process and analyse the data is available in the scripting language python under the github repository https://github.com/pohleric/toe_tools.

**Data availability**

20th Century Reanalysis V2c data provided by the NOAA/OAR/ESRL PSD, Boulder, Colorado, USA, from their website at http://www.esrl.noaa.gov/psd/. The CRUNCEP Version 7 data is available through registration following the website https://rda.ucar.edu/datasets/ds314.3/. ERA20 data are available from ECMWF Data Servers through the python module 'ecmwfapi' https://pypi.org/project/ecmwf-api-client/. The RIHMI observational dataset used in this study can be obtained through the website https://cdiac.ess-dive.lbl.gov/ndps/russia_daily518.html.

**Video supplement**

Video1 – Spatiotemporal evolution of emergence for temperature in the Lena River catchment for the different data sources (by row): 1) CRUNCEP, 2) average emergence of all individual CS, 3) average of the HD of the 10 best CS, and 4) the single best performing model to showcase the higher variability of

individual models compared to the averaged evolutions. Columns from left to right represent the different temporal analyses annual, summer, winter. Blue dots indicate a negative sign of the emergence.

Video2 – Same as Video1 but for precipitation.

## Supplement

The supplement is added as additional document and provides information about the spatiotemporal variability of datasets and gives a more detailed view on some statistics.

## Author contribution

EP and CG designed the study; EP did the calculation and produced the figures, maps, and the toolbox; EP wrote the outline and all authors contributed to the discussion and refinement of the manuscript.

## Competing Interests

The authors declare that they have no conflict of interest.

## Acknowledgements

This work benefited from the French state aid managed by the ANR under the "Investissements d'avenir" programme with the reference ANR-11-IDEX-0004 - 17-EURE-0006. We thank the IPSL-EUR postdoc
initiative that initiated a workshop on the ToE issue. In the context of this workshop we particularly acknowledge the discussions with Pascal Terray, Goulven Laruelle, Marco Gaetani, Vincent Thieu. Special thanks go to Alexander Fedorov and Pavel Konstantinov from the Melnikov Permafrost Institute, Yakutsk, for providing data and discussion. We acknowledge the World Climate Research Program's Working Group on Coupled Modelling, which is responsible for CMIP, and we thank the climate
modeling groups (listed in Table S1 of this paper) for producing and making available their model output. For CMIP the U.S. Department of Energy's Program for Climate Model Diagnosis and Intercomparison provides coordinating support and led development of software infrastructure in partnership with the Global Organization for Earth System Science Portals. Support for the Twentieth Century Reanalysis Project dataset is provided by the U.S. Department of Energy, Office of Science Innovative and Novel
Computational Impact on Theory and Experiment (DOE INCITE) program, and Office of Biological and Environmental Research (BER), and by the National Oceanic and Atmospheric Administration Climate Program Office.

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

**Table 1: Nash-Sutcliffe efficiency statistics of the 10 best climate simulations with respect to the CRUNCEP data for each pixel encompassing a meteorological station with records of more than 10 years in the 1901-1921 reference period. Positive NSE in bold.**

| station | Kirensk | Olekminsk | Ust'-Maja | Viljujsk | Yakutsk |
|---|---|---|---|---|---|
| ***Temperature*** | | | | | |
| **annual** | | | | | |
| NSE_mean | **0.58** | **0.58** | **0.61** | **0.43** | **0.62** |
| NSE_max | **0.79** | **0.81** | **0.74** | **0.74** | **0.73** |
| NSE_min | **0.44** | **0.43** | **0.53** | **0.26** | **0.53** |
| **summer** | | | | | |
| NSE_mean | **0.27** | **0.35** | -0.02 | **0.29** | **0.07** |
| NSE_max | **0.55** | **0.49** | **0.22** | **0.70** | **0.30** |
| NSE_min | **0.13** | **0.14** | -0.13 | **0.01** | -0.08 |
| **winter** | | | | | |
| NSE_mean | **0.21** | **0.57** | **0.35** | -0.04 | **0.38** |
| NSE_max | **0.46** | **0.71** | **0.62** | **0.44** | **0.59** |
| NSE_min | -0.08 | **0.47** | **0.19** | -0.68 | **0.29** |
| ***Precipitation*** | | | | | |
| **annual** | | | | | |
| NSE_mean | **0.10** | -0.68 | -1.32 | -1.09 | -0.89 |
| NSE_max | **0.46** | **0.13** | -0.48 | 0.00 | -0.19 |
| NSE_min | -0.12 | -1.18 | -1.80 | -1.94 | -1.58 |
| **summer** | | | | | |
| NSE_mean | -0.20 | -0.91 | -0.50 | -1.86 | -1.18 |
| NSE_max | **0.36** | **0.22** | **0.39** | -0.25 | **0.24** |
| NSE_min | -0.56 | -1.66 | -0.95 | -3.37 | -2.40 |
| **winter** | | | | | |
| NSE_mean | -19.49 | **0.07** | -16.88 | -0.07 | **0.24** |
| NSE_max | -1.36 | **0.40** | -9.10 | **0.20** | **0.54** |
| NSE_min | -31.83 | -0.12 | -20.60 | -0.19 | **0.08** |

**Table 2: Area-wide ToE based on the mean HD of all 65 CMIP5 model simulations and the corresponding change in temperature or precipitation at different emergence levels (HD) and seasons.**

|                 | ToE [year] |      |      | Change [ºC($T$) or mm($P$)] |       |       |
| --------------- | ---------- | ---- | ---- | --------------------------- | ----- | ----- |
| Emergence level | 30%        | 40%  | 50%  | 30%                         | 40%   | 50%   |
| T (annual)      | 1981       | 1992 | 2001 | 0.75                        | 1.11  | 1.48  |
| T (summer)      | 1992       | 2005 | 2016 | 0.83                        | 1.19  | 1.57  |
| T (winter)      | 1991       | 2004 | 2015 | 1.48                        | 2.12  | 2.77  |
| P (annual)      | 2034       | 2049 | 2061 | 49.08                       | 73.38 | 98.27 |
| P (summer)      | 2055       | 2067 | 2073 | 46.16                       | 66.80 | 87.99 |
| P (winter)      | 2026       | 2041 | 2053 | 14.77                       | 21.68 | 28.99 |