# Peer review of "Emerging climate signals in the Lena River catchment: a nonparametric statistical approach"

_Hydrology and Earth System Sciences, 2019_

## Referee Comment (RC1) · Wolfgang Schwanghart (Referee) · 4 Oct 2019

Pohl et al. derive a new metric that enables measuring the time-of-emergence (ToE) of time series. They apply the method to temporal-spatial climate data for the Lena River catchment. In general, the paper is well written and easy to follow. The figures have a high quality although they are very difficult to read in grayscale prints. A more printer-friendly colorscheme might be chosen by the authors when revising their manuscript.

Major comments

The study aims to present a novel ToE approach but the paper mainly focuses on the application of the ToE rather than its evaluation. An in-depth evaluation would be necessary to enable readers to acknowledge the benefits and shortcomings of a new

metric. A possible solution is to include an extra section that evaluates the sensitivity and power of the method using simulated data.

It remains shrouded whether the authors use a kernel density estimator (KDE) that enables comparing the pdfs. Looking at the code (toe_calc.py), the authors seem to use a gaussian KDE, but this is not shown in the manuscript. As it stands, the HD is calculated for discrete probability distributions. If a KDE approach is actually chosen, than kernel bandwidth is an additional hyperparameter that most software choose automatically but that should be controlled.

Finally, I am not entirely convinced whether the HD-based ToE approach presents a sufficiently sophisticated new technique. Uncertainties in the metric have mainly been addressed using different climate models. However, there are other uncertainties that are not sufficiently captured by the method. First, each pdf presents a sample itself which is subject to uncertainties. This uncertainty is related to the window width. As a consequence, the HD-based ToE is a stochastic variable, which is prone to uncertainty. I think that this uncertainty is not sufficiently addressed by the authors, although it is recognized (17.14f).

In summary, I think that the paper needs major revisions to address some shortcomings in uncertainty quantification and evaluation of the HD-based ToE. Moreover, the possible impact of the KDE bandwidth on the results should be assessed.

Minor comments

5.29 - The equation lacks a term on the right (e.g HD(Q,R) = ). In addition, I don't understand how you obtain the PDFs from the data. Do you use a kernel density approach? Otherwise, the equation is valid for discrete probability distributions only.

6.27 - Again, if a kernel density approach is used, then there are other parameters that include the type of kernel (gaussian, triangle, ...) and the bandwidth. These parameters should be kept constant if pdfs are compared. Automatic bandwidth determination is

challenging if you have skewed, multimodal, and bounded data (e.g. only positive data such as precipitation).

7.4 - The coefficient of determination (r2) is actually a poor measure because it only evaluates the linear fit between the datasets. However, it may be more interesting whether the models capture the means and variability correctly and thus, the Nash-Sutcliffe efficiency (NSE) as used later may be a better choice.

9.12 - Is it possible that CRUNCEP was actually derived using empirical data from these stations? That would explain the very good correlation. This would also explain that areas far from stations show these artifacts (9.19). You may discuss this in more detail in section 5.3.

15.7 - basically non-parametric -> remove basically.

16.43 - avoid qualitative statements such as "huge"

19.37 - These other ToE methods relying on thresholds or statistial test actually rely on continuous metrics, too. Thresholds are derived from some metric, e.g. max. distance between two distributions such as the KS-test, and tests often rely on p-values which are continuous, too. In this respect, the HD-based ToE is not much different.

Fig.7 - The colorbar does not allow distinguishing regions that have years of emergence in 1960 or in 2088.

---

## Referee Comment (RC2) · Anonymous Referee #2 · 24 Oct 2019

This is an interesting paper which presents a new approach for detecting emergence of climate change, applied to a basin in eastern Siberia. The approach looks justified and offers some important advantages over more conventional approaches. I have attached my marked copy of the manuscript with several grammatical corrections, but otherwise should be accepted.

Please also note the supplement to this comment:
https://www.hydrol-earth-syst-sci-discuss.net/hess-2019-360/hess-2019-360-RC2-supplement.pdf
* * *
[Figure]

**Supplement:**

[revised manuscript text omitted]

---

## Author Comment (AC1) · 31 Jan 2020

We would like to thank the reviewer for the thorough reading and relevant comments and suggestions. Below are our point-by-point author comments (ACs) to the referee comments (RCs). We provide an additional three figures to answer the comments. The figure captions are at the end of this text due to limitations in the submission website.

Referee 1:

RC1: The study aims to present a novel ToE approach but the paper mainly focuses on the application of the ToE rather than its evaluation. An in-depth evaluation would be necessary to enable readers to acknowledge the benefits and shortcomings of a new metric. A possible solution is to include an extra section that evaluates the sensitivity

and power of the method using simulated data.

AC1: We agree that the introduction of the method and a more detailed sensitivity analysis has fallen short. In combination with RC2, RC3, we propose to follow the suggestion of the referee to include an additional section on benefits, shortcomings, and sensitivity of the novel ToE approach. We will do this based on synthetically generated data, where we control onset and strength in signal changes. We first use two datasets, one closely resembling a temperature time-series of the used climate model datasets (type 1), and one that serves to showcase detection sensitivity (type 2). We calculate the Hellinger distance (HD) based on a KDE-PDF (see also RC2/AC2) and compare the results against the Kolmogorov-Smirnov metric (KS), which is the maximum distance between the two cumulative density functions of the data to be compared.

The type 1 and type 2 data are normal distributed data with a fixed mean and standard deviation (SD) until the breakpoint year (1960). Thereafter, type 1 data have a fixed linear change (slope derived from arbitrarily chosen pixel of one of the climate model simulations), while the SD stays constant. The type 2 data have a constant mean value after the breakpoint year, but a continuous increase in standard deviation, reaching two times the reference SD at the end of the time-series. We generate 5000 time-series of each type of dataset and calculate the HD and KS. Figure 1 showcases one representation of each of the synthetic time-series (upper panels). The distance plots (lower panels) show the median (bold line), inner quartile range (shading), and the 5%-95% percentiles (points) to give a representative assessment of how the two distance metrics perform.

Generally speaking, the HD has a crucial advantage over KS in terms of continuous change description and also in terms of accuracy. The left panel in Figure 1 shows the co-evolution of HD and KS for a time series with a pronounced trend. A step-function like evolution is visible for KS. This becomes even clearer if the change in the original signal is smaller (Figure 1 – right). The inner quartile range (IQR) of the HD based on 5000 samples of the time-series is mostly lower than for KS. Also, the 90% range

(5%-95% percentile) reaches overall lower values compared to the KS, as well as less variability along the time axis (the KS changes from a low to a high range within a few years). The right panels in Figure 1 show a signal with slight changes (gradual increase of the standard deviation) and corresponding distances. The KS is not able to detect the change in a continuous way and only indicates change once a certain threshold is crossed. The accuracy, i.e. the range in distance estimates based on the 5000 samples, is very similar in this case. The step-function-like evolution in KS is depending on the sample size, which determines the minimum dissimilarity increase (1/n, with n being the sample size). This step-function like evolution is also clearly visible in the example with the strong onset of a trend (left). Not shown are the minimum and maximum values that are possible. For KS, these have a wider range because even a very slight shift of an otherwise equal distribution can cause a high KS. However, this does not happen often (not captured by the 90% range).

Figure 2 shows the effect of the KDE kernel bandwidth. This is a hyperparameter that is either handled automatically or through a defined function. In our approach we use an automatic bandwidth estimation that is based on Scotts's factor (Scott, 2015), which is only dependent on the number of data points and dimensionality of the data (See Figure 2). As such, it is fixed in the present work for a fixed window width and one-dimensional time-series. For assumed common sample sizes for monthly to annual data between 10 and 100, Scott's factor provides bandwidths between 0.4 and 0.6. The resulting change in HD is shown in Figure 2. We use again the two synthetic time-series from before to show the change in HD. For the relatively large range in sample sizes and resulting change in bandwidths, the overall change is in the range of only 5%. Even though it could be interesting to investigate the different effects of hyperparameters further, also including the shape of kernels (see also AC5), this is out of the scope of the present work.

Figure 3 showcases the evolution in HD for strictly positive data with an example of a gamma distribution and different shape parameters. Because the KDE approach is
[Figure]

fitting symmetrical kernels, the approach must introduce some uncertainty if the true distribution is strictly positive. Therefore, in Figure 3 we calculated the HDs based on the actual PDFs of gamma distributions with different shape parameters (x-axis). We then compare these HDs with the HDs based on the KDE approach (y-axis). We again showcase the effect of different bandwidths (by using according sample sizes) on the outcome. Figure 3 demonstrates that for real HDs of lower than 30%, there is a pronounced positive bias of the KDE approach. This bias occurs not only for HDs between distributions with small shape parameters, i.e. distributions with a pronounced peak near 0, but also when distributions have their mean far away from 0. The bias is gradually reduced for a bigger sample size and bandwidth. For larger HD (true and estimated), distances using different sample sizes are in close agreement. The fact that the difference between HD of true and estimated PDFs are also in close agreement for small shape parameters supports possible application for non-normal distributed data, e.g. precipitation data at high temporal resolution.

We propose to discuss the addressed issues in the here presented way. But we also think that a more detailed analysis should be done in a separate study.

RC2: It remains shrouded whether the authors use a kernel density estimator (KDE) that enables comparing the pdfs. Looking at the code (toe_calc.py), the authors seem to use a gaussian KDE, but this is not shown in the manuscript. As it stands, the HD is calculated for discrete probability distributions. If a KDE approach is actually chosen, than kernel bandwidth is an additional hyperparameter that most software choose automatically but that should be controlled.

AC2: We use the discrete formulation to calculate the HD. But we indeed use a KDE to estimate the PDF. In order to use the formula presented, we evaluate the PDF along a series of x-values (n=200, as described on page 6). We have not discussed the effect of the bandwidth selection on the HD outcome. As also mentioned in AC1, the impact of bandwidth on the HD is relatively low (Figure 2). This might result from the implementation of Scott's factor in the KDE approach we used. The bandwidth is

varying not as much as it would when using other bandwidth estimators. As we show in Figure 2, even if sample sizes vary between 10 and 100, the resulting HD would only differ by around 5%. We do agree, however, that this needs to be made clear in the manuscript and we added these pieces of information. As this is an important aspect, we have added an option in the code to keep the bandwidth fixed, independent of the sample size.

RC3: Finally, I am not entirely convinced whether the HD-based ToE approach presents a sufficiently sophisticated new technique. Uncertainties in the metric have mainly been addressed using different climate models. However, there are other uncertainties that are not sufficiently captured by the method. First, each pdf presents a sample itself which is subject to uncertainties. This uncertainty is related to the window width. As a consequence, the HD-based ToE is a stochastic variable, which is prone to uncertainty.I think that this uncertainty is not sufficiently addressed by the authors, although it is recognized (17.14f).

In summary, I think that the paper needs major revisions to address some shortcomings in uncertainty quantification and evaluation of the HD-based ToE. Moreover, the possible impact of the KDE bandwidth on the results should be assessed.

AC3: An estimated PDF, based on a somehow arbitrarily chosen window sizes is of course introducing uncertainties. But it should also be clear that we aim particularly at providing a technique that addresses a much bigger uncertainty, namely the definition of a background variability using randomly chosen parameterizations. In previous works, addressing uncertainties through the choice of meta-parameters has often fallen short. We provide a sensitivity analysis in which we test systematically various possible window sizes (all meaningful ones in the present setup) and time-series splitting points ("split year"). It is not feasible to test all possible combinations of hyper-parameters and datasets and present them within the chosen scope of work. We think that this would be a good idea in another study, where the focus could be strictly set to algorithm inter-comparison and sensitivity. In the present work, we particularly aimed at

looking at Eastern Siberia for its representativeness of northern latitude permafrost landscapes and its potential huge importance in the climate system. By using different datasets and showcasing how the various challenges related to data uncertainty impact ToE estimates, the sensitivity analysis of the actual method has become a bit shorter. We hope that, with the additional figures and tests, we can provide enough evidences that our method provides a novel approach with advantages (despite the disadvantages from the hyper-parameters) to justify the publication of the work after including the proposed changes and additions.

Minor comments

RC4: 5.29 - The equation lacks a term on the right (e.g HD(Q,R) = ). In addition, I don't understand how you obtain the PDFs from the data. Do you use a kernel density approach? Otherwise, the equation is valid for discrete probability distributions only.

AC4: We indeed use a KDE. We added the missing information as described before and adjusted the equation to include a term on the right.

RC5: 6.27 - Again, if a kernel density approach is used, then there are other parameters that include the type of kernel (gaussian, triangle, ...) and the bandwidth. These parameters should be kept constant if pdfs are compared. Automatic bandwidth determination is challenging if you have skewed, multimodal, and bounded data (e.g. only positive data such as precipitation).

AC5: As commented before, we have indeed missed to address this problem. We hope to provide sufficient evidence with Figure 2, Figure 3, and the new section that the choice in bandwidth has actually a rather small impact on the outcome, independent of the change in the time-series (comparing the two synthetic examples). Moreover, for this study we used a fixed window width of 21 years. Because Scott's factor is used to determine the bandwidth, and because this is only dependent on the sample size (n=21) and dimension (constant), the bandwidth is kept constant in the study. As we show in Figure 2 and Figure 3, even if we would have used varying sample sizes, the

expected changes would be in a range far lower than the uncertainties arising from different climate model simulations. We have not tested the effect of kernel types because these are widely believed to be of minor importance compared to the bandwidth selection (e.g. Turlach, 1993).

RC6: 7.4 - The coefficient of determination (r2) is actually a poor measure because it only evaluates the linear fit between the datasets. However, it may be more interesting whether the models capture the means and variability correctly and thus, the Nash-Sutcliffe efficiency (NSE) as used later may be a better choice.

AC6: This initial test serves the purpose to find the gridded dataset that provides the most realistic values for both temperature and precipitation with respect to the observational data. While a metric like the NSE could provide a more detailed assessment, r2 in combination with the figure is in our opinion sufficient to show that all but the CRUNCEP data do not realistically represent the precipitation records (see in particular the precipitation scatter plots in Figure 3 in the manuscript). Therefore, the r2 (in combination with the figure) suffices the purpose to find the "best" available dataset.

RC7: 9.12 - Is it possible that CRUNCEP was actually derived using empirical data from these stations? That would explain the very good correlation. This would also explain that areas far from stations show these artifacts (9.19). You may discuss this in more detail in section 5.3.

AC7: The CRUNCEP is indeed based on observational data. This was not sufficiently described in Section 3.2.1. It was mentioned that the data results from the CRU TS v3.24 monthly climate dataset but without pointing out that this dataset incorporates observational data. We will make this point clear. We will repeat this information in the results section (9.18), where the artificial behavior is indeed directly related to that, and as suggested in section 5.3.

RC8: 15.7 - basically non-parametric -> remove basically.

AC8: We will thoroughly check the language with a native speaker again to improve the language.

RC9: 16.43 - avoid qualitative statements such as "huge"

AC9: We will check the entire text again to change qualitative to quantitative statements where possible, or remove the qualitative statements.

RC10: 19.37 - These other ToE methods relying on thresholds or statistial test actually rely on continuous metrics, too. Thresholds are derived from some metric, e.g. max. distance between two distributions such as the KS-test, and tests often rely on p-values which are continuous, too. In this respect, the HD-based ToE is not much different.

AC10: The KS-test is the only test for ToE applications, that we are aware of, that uses a distance metric based on the data distributions. Even in these cases the distance metric is always used in combination with a fixed significance level, which is described in Section 1.1 and Section 5.1. Emergence is thus not represented as a continuous metric. We will point this out more clearly in the two relevant sections, as well as in the newly added section (AC1). Some relevant differences between the HD and KS metric (if it was used instead) are highlighted in the new section (AC1). It should be noted that not the KS metric (AC1), but the KS test with predefined significance levels is used in previous studies. Furthermore, our approach provides the benefit to directly compare the emergence of observational data and climate model simulations that can aid a selection of more suitable simulations. This is not possible using previous methods (Section 5.1).

RC11: Fig.7 - The colorbar does not allow distinguishing regions that have years of emergence in 1960 or in 2088.

AC11: We will remove the first color, so that all colors will be unique.

References

Scott, D. W. (2015). Multivariate density estimation: theory, practice, and visualization.

John Wiley & Sons.

Turlach, B. A. (1993). Bandwidth selection in kernel density estimation: A review. CORE and Institut de Statistique.

Figure Captions

Figure 1: Comparison between Hellinger distance (HD) and the Kolmogorov-Smirnov metric (KS). Two synthetic time-series examples (top) and corresponding distance evolution (bottom). Inner quartile range (IQR) and 5%-95% percentile range (middle). Note the different scales.

Figure 2: HD sensitivity to KDE-bandwidth. Automatic bandwidth selection in python's scipy.stats KDE is based on Scott's factor (Scott, 2015), where n is the sample size and d is the dimension. For 1-dimensional data, sample sizes between 10 and 100 correspond to bandwidths of 0.65 to 0.4. Middle panel is sensitivity in HD for example 1 (Figure 1 – left-hand side). Right panel is sensitivity in HD for example 2 (Figure 1 right-hand side) at years 1940, 2000, and 2050.

Figure 3: Comparison of Hellinger distances for strictly positive data (by means of a gamma distribution) calculated from the actual PDF vs. the KDE-PDF approach as used in the manuscript. The different sample sizes for the KDE-PDFs correspond roughly to the different bandwidths tested in Figure 2. Inset shows five of the 16 gamma distributions with different shape parameters "a" between 0.2 and 5.0 used for this test. The HD between each possible gamma distribution with a different shape parameter was calculated. This was done for both, the "true" (the actual PDFs) and the estimated (KDE) PDFs. This was repeated 100 times for each distance. The averages of these distances are plotted. The grey-coding of points (face color) represents the difference in shape parameter a, with light colors for small and dark colors for large differences.

[Figure]

**Fig. 1.** (too long; see section Figure Captions)

[Figure]

**Fig. 2.** (too long; see section Figure Captions)

**Gamma distribution distances**

HD distances (KDE-PDFs)

- a=0.2
- a=0.84
- a=1.48
- a=2.76
- a=5.0

- ○ KDE-PDF (n=11)
- ○ KDE-PDF (n=21)
- ○ KDE-PDF (n=99)

HD distances (true PDFs)

**Fig. 3.** (too long; see section Figure Captions)

---

## Author Comment (AC2) · 31 Jan 2020

We would like to thank the reviewer for the thorough reading and relevant comments and suggestions. Below is our author comment (AC) to the referee comments (RC).

Referee 2:

RC: This is an interesting paper which presents a new approach for detecting emergence of climate change, applied to a basin in eastern Siberia. The approach looks justified and offers some important advantages over more conventional approaches. I have attached my marked copy of the manuscript with several grammatical corrections, but otherwise should be accepted.

AC: We thank the referee for the grammatical corrections and incorporate these into

the updated manuscript.

---

## Author Response (AR1)

**Point-by-point response letter for "Emerging climate signals in the Lena River catchment: a non-parametric statistical approach"**

Eric Pohl[1], Christophe Grenier[1], Mathieu Vrac[1], Masa Kageyama[1]

[1]Laboratoire des Sciences du Climat et de l'Environnement (LSCE/IPSL), UMR CEA-CNRS-UVSQ, Gif-sur-Yvette, 91120, France

*Correspondence to*: Eric Pohl (Eric.Pohl@lsce.ipsl.fr)

We are happy to submit our revised manuscript. We hope to have satisfactorily addressed all raised issues by the two referees. The following text lists again the point-by-point responses to the reviewer comments and indicates where in the manuscript we have made according changes. A manuscript highlighting the changes in comparison to the previous version is attached as well.

Referee 1:

**RC1:**
The study aims to present a novel ToE approach but the paper mainly focuses on the application of the ToE rather than its evaluation. An in-depth evaluation would be necessary to enable readers to acknowledge the benefits and shortcomings of a new metric. A possible solution is to include an extra section that evaluates the sensitivity and power of the method using simulated data.

**AC1:**
We agree that the introduction of the method and a more detailed sensitivity analysis has fallen short. We have introduced a clarification on the theoretically continuous distance character of the KS-metric within the KS.
P3 L11-13
P6 L3-10

In combination with **RC2, RC3**, we have included an additional section (Section 4) on benefits, shortcomings, and sensitivity of the KDE-PDF and resulting HD.
P6 L49-50
P7 L1-4
P8 L28 – P10 L2
New figures: Fig. 3, Fig. 4, Fig. S2

For this, we have generated synthetic data with a controlled onset and strength in signal changes. The datasets closely resembling a temperature time-series of the used climate model datasets (type 1), and a time-series that serves to showcase detection sensitivity (type 2). We calculate the Hellinger distance (HD) based on a KDE-PDF and compare the results against the Kolmogorov-Smirnov metric (KS-metric), which is the maximum distance between the two cumulative density functions of the data to be compared.

The type 1 and type 2 data are normal distributed data with a fixed mean and standard deviation (SD) until the breakpoint year (1960). Thereafter, type 1 data have a fixed linear change (slope derived from arbitrarily chosen pixel of one of the climate model simulations), while the SD stays constant. The type 2 data have a constant mean value after the breakpoint year, but a continuous increase in standard deviation, reaching two times the reference SD at the end of the time-series. We generate 5000 timeseries of each type of dataset and calculate the HD and KS. Figure 3 (in the manuscript) showcases one representation of each of the synthetic time-series (upper panels). The distance plots (lower panels) show the median (bold line), inner quartile range (shading), and the 5%-95% percentiles (points) to give a representative assessment of how the two distance metrics perform.

Generally speaking, the HD has a crucial advantage over KS in terms of continuous change description and also in terms of accuracy. The left panel in Figure 3 shows the co-evolution of HD and KS for a time series with a pronounced trend. A step-function like evolution is visible for KS. This becomes even clearer if the change in the original signal is smaller (Figure 3 – right). The inner quartile range (IQR) of
10 the HD based on 5000 samples of the time-series is mostly lower than for KS. Also, the 90% range (5%-95% percentile) reaches overall lower values compared to the KS, as well as less variability along the time axis (the KS changes from a low to a high range within a few years).

The right panels in Figure 3 show a signal with slight changes (gradual increase of the standard deviation) and corresponding distances. The KS is not able to detect the change in a continuous way and
15 only indicates change once a certain threshold is crossed. The accuracy, i.e. the range in distance estimates based on the 5000 samples, is very similar in this case. The step-function-like evolution in KS is depending on the sample size, which determines the minimum dissimilarity increase ($1/n$, with $n$ being the sample size). This step-function like evolution is also clearly visible in the example with the strong onset of a trend (left). Not shown are the minimum and maximum values that are possible. For
20 KS, these have a wider range because even a very slight shift of an otherwise equal distribution can cause a high KS. However, this does not happen often (not captured by the 90% range).

Figure 4 shows the effect of the KDE kernel bandwidth. This is a meta-parameter that is either handled automatically or through a defined function. In our approach we use an automatic bandwidth estimation
25 that is based on Scotts's factor (Scott, 2015), which is only dependent on the number of data points and dimensionality of the data (see Figure 4). As such, it is fixed in the present work for a fixed window width and one-dimensional time-series. For assumed common sample sizes for monthly to annual data between 10 and 100, Scott's factor provides bandwidths between 0.4 and 0.6. The resulting change in HD is shown in Figure 4. We use again the two synthetic time-series from before to show the change in
30 HD. For the relatively large range in sample sizes and resulting change in bandwidths, the overall change is in the range of only 5%. A more excessive analysis on how the bandwidth affects different types of signals is out of scope for this work. We have not tested the effect of kernel types because these are widely believed to be of minor importance compared to the bandwidth selection (e.g. Turlach 1993, Bianchi, 1995). However, we also tested the impact of strictly positive and strongly skewed
35 distributions on the approach using Gamma distributions with different shape parameters (Supplementary – Fig. S2).

Figure S2 showcases the evolution in HD for strictly positive data with an example of a Gamma distribution and different shape parameters. Because the KDE approach is fitting symmetrical kernels,
40 the approach must introduce some uncertainty if the true distribution is strictly positive. Therefore, in Figure 3 we calculated the HDs based on the actual PDFs of gamma distributions with different shape parameters (x-axis). We then compare these HDs with the HDs based on the KDE approach (y-axis). We again showcase the effect of different bandwidths (by using according sample sizes) on the outcome. Figure 3 demonstrates that for real HDs of lower than 30%, there is a pronounced positive
45 bias of the KDE approach. This bias occurs not only for HDs between distributions with small shape parameters, i.e. distributions with a pronounced peak near 0, but also when distributions have their mean far away from 0. The bias is gradually reduced for a bigger sample size and bandwidth. For larger HD (true and estimated), distances using different sample sizes are in close agreement. The fact that the difference between HD of true and estimated PDFs are also in close agreement for small shape
50 parameters supports possible application for non-normal distributed data, e.g. precipitation data at high temporal resolution.

We have also included an additional paragraph in the Discussion to reflect the additional tests and thoughts, and included a sentence in the conclusions.

**RC2:**

It remains shrouded whether the authors use a kernel density estimator (KDE) that enables comparing the pdfs. Looking at the code (toe_calc.py), the authors seem to use a gaussian KDE, but this is not shown in the manuscript. As it stands, the HD is calculated for discrete probability distributions. If a KDE approach is actually chosen, than kernel bandwidth is an additional meta-parameter that most software choose automatically but that should be controlled.

**AC2:**

We use the discrete formulation to calculate the HD. But we indeed use a KDE to estimate the PDF. In order to use the formula presented, we evaluate the PDF along a series of x-values (n=200, as described on page 6).

We have not discussed the effect of the bandwidth selection on the HD outcome. As mentioned in **AC1**, the impact of bandwidth on the HD is relatively low (Figure 2). This might result from the implementation of Scott's factor in the KDE approach we used. The bandwidth is varying not as much as it would when using other bandwidth estimators. As we show in Figure 4, even if sample sizes vary between 10 and 100, the resulting HD would only differ by around 5%. We do agree, however, that this needs to be made clear in the manuscript and we added these pieces of information. As this is an important aspect, we have added an option in the code to keep the bandwidth fixed, independent of the sample size.

As mentioned in AC1, we have included this in the Discussion.

**RC3:**

Finally, I am not entirely convinced whether the HD-based ToE approach presents a sufficiently sophisticated new technique. Uncertainties in the metric have mainly been addressed using different climate models. However, there are other uncertainties that are not sufficiently captured by the method. First, each pdf presents a sample itself which is subject to uncertainties. This uncertainty is related to the window width. As a consequence, the HD-based ToE is a stochastic variable, which is prone to uncertainty. I think that this uncertainty is not sufficiently addressed by the authors, although it is recognized (17.14f).

In summary, I think that the paper needs major revisions to address some shortcomings in uncertainty quantification and evaluation of the HD-based ToE. Moreover, the possible impact of the KDE bandwidth on the results should be assessed.

**AC3:**

An estimated PDF, based on a somehow arbitrarily chosen window sizes is of course introducing uncertainties. But it should also be clear that we aim particularly at providing a technique that addresses a much bigger uncertainty, namely the definition of a background variability using randomly chosen parameterizations. In previous works, addressing uncertainties through the choice of meta-parameters has often fallen short.

We provide a sensitivity analysis in which we test systematically various possible window sizes (all meaningful ones in the present setup) and time-series splitting points ("split year"). It is not feasible to test all possible combinations of meta-parameters and datasets and present them within the chosen scope of work. We think that this would be a good idea in another study, where the focus could be strictly set to algorithm inter-comparison and sensitivity.

In the present work, we particularly aimed at looking at Eastern Siberia for its representativeness of northern latitude permafrost landscapes and its potential huge importance in the climate system. By using different datasets and showcasing how the various challenges related to data uncertainty impact ToE estimates, the sensitivity analysis of the actual method has become a bit shorter.

5   We hope that, with the additional figures and tests, we can provide enough evidences that our method provides a novel approach with advantages (despite the disadvantages from the meta-parameters) to justify the publication of the work after including the proposed changes and additions.

[Figure]

**Figure 1: Comparison between Hellinger distance (HD) and the Kolmogorov-Smirnov metric (KS). Two synthetic time-series examples (top) and corresponding distance evolution (bottom). Inner quartile range (IQR) and 5%-95% percentile range (middle).**
15   **Note the different scales.**

[Figure]

**Figure 2: HD sensitivity to KDE-bandwidth. Automatic bandwidth selection in python's scipy.stats KDE is based on Scott's factor (Scott, 2015), where n is the sample size and d is the dimension. For 1-dimensional data, sample sizes between 10 and 100 correspond to bandwidths of 0.65 to 0.4. Middle panel is sensitivity in HD for example 1 (Figure 1 – left-hand side). Right panel is**
20   **sensitivity in HD for example 2 (Figure 1 right-hand side) at years 1940, 2000, and 2050.**

[Figure]

**Figure 3: Comparison of Hellinger distances for strictly positive data (by means of a gamma distribution) calculated from the actual PDF vs. the KDE-PDF approach as used in the manuscript. The different sample sizes for the KDE-PDFs correspond roughly to the different bandwidths tested in Figure 2. Inset shows five of the 16 gamma distributions with different shape parameters "a" between 0.2 and 5.0 used for this test. The HD between each possible gamma distribution with a different shape parameter was calculated. This was done for both, the "true" (the actual PDFs) and the estimated (KDE) PDFs. This was repeated 100 times for each distance. The averages of these distances are plotted. The grey-coding of points (face color) represents the difference in shape parameter a, with light colors for small and dark colors for large differences.**

Minor comments

**RC4:**

5.29 - The equation lacks a term on the right (e.g HD(Q,R) = ). In addition, I don't understand how you obtain the PDFs from the data. Do you use a kernel density approach? Otherwise, the equation is valid for discrete probability distributions only.

**AC4:**

We indeed use a KDE. We added the missing information as described before and adjusted the equation to include a term on the right.

**RC5:**

6.27 - Again, if a kernel density approach is used, then there are other parameters that include the type of kernel (gaussian, triangle, ...) and the bandwidth. These parameters should be kept constant if pdfs are compared. Automatic bandwidth determination is challenging if you have skewed, multimodal, and bounded data (e.g. only positive data such as precipitation).

**AC5:**

As commented before, we have indeed missed to address this problem. We hope to provide sufficient evidence with Figure 2, Figure 3, and the new section that the choice in bandwidth has actually a rather small impact on the outcome, independent of the change in the time-series (comparing the two synthetic examples). Moreover, for this study we used a fixed window width of 21 years. Because Scott's factor is used to determine the bandwidth, and because this is only dependent on the sample size (n=21) and dimension (constant), the bandwidth is kept constant in the study. As we show in Figure 2 and Figure 3, even if we would have used varying sample sizes, the expected changes would be in a range far lower than the uncertainties arising from different climate model simulations. We have not tested the effect of

kernel types because these are widely believed to be of minor importance compared to the bandwidth selection (e.g. Turlach, 1993).

**RC6:**
7.4 - The coefficient of determination (r2) is actually a poor measure because it only evaluates the linear fit between the datasets. However, it may be more interesting whether the models capture the means and variability correctly and thus, the Nash- Sutcliffe efficiency (NSE) as used later may be a better choice.

**AC6:**
This initial test serves the purpose to find the gridded dataset that provides the most realistic values for both temperature and precipitation with respect to the observational data. While a metric like the NSE could provide a more detailed assessment, r2 in combination with the figure is in our opinion sufficient to show that all but the CRUNCEP data do not realistically represent the precipitation records (see in particular the precipitation scatter plots in Figure 5 in the manuscript). Therefore, the r2 (in combination with the figure) suffices the purpose to find the "best" available dataset.

**RC7:**
9.12 - Is it possible that CRUNCEP was actually derived using empirical data from these stations? That would explain the very good correlation. This would also explain that areas far from stations show these artifacts (9.19). You may discuss this in more detail in section 5.3.

**AC7:**
The CRUNCEP is indeed based on observational data. This was not sufficiently described in Section 3.2.1. It was mentioned that the data results from the CRU TS v3.24 monthly climate dataset but without pointing out that this dataset incorporates observational data. We have included this information in section 3.2.1, in the results section (P10 L22-23), where the artificial behavior is indeed directly related to that, and as suggested in section 6.3.

P8 L6-8
P10 L17-18
P19 L40-45

**RC8:**
15.7 - basically non-parametric -> remove basically.

**AC8:**
Has been removed and the manuscript has been proof-read again.

**RC9:**
16.43 - avoid qualitative statements such as "huge"

**AC9:**
We changed qualitative to quantitative statements where possible, removed some qualitative statements, or added quantities to provide reasoning for the qualitative statements. Some of these statements remain, e.g. if a visual pattern is described.

**RC10:**
19.37 - These other ToE methods relying on thresholds or statistial test actually rely on continuous metrics, too. Thresholds are derived from some metric, e.g. max. distance between two distributions

such as the KS-test, and tests often rely on p-values which are continuous, too. In this respect, the HD-based ToE is not much different.

**AC10:**
The KS-test is the only test for ToE applications, that we are aware of, that uses a distance metric based on the data distributions. Even in these cases the distance metric is always used in combination with a fixed significance level, which is described in Section 1.1 and Section 6.1. Emergence is thus not represented as a continuous metric. We will point this out more clearly in the two relevant sections, as well as in the newly added section (AC1). Some relevant differences between the HD and KS metric (if it was used instead) are highlighted in the new section (AC1). It should be noted that not the KS metric (AC1), but the KS test with predefined significance levels is used in previous studies.
Furthermore, our approach provides the benefit to directly compare the emergence of observational data and climate model simulations that can aid a selection of more suitable simulations. This is not possible using previous methods (Section 6.1).

P3 L11-13
P18 L26-30

**RC11:**
Fig.7 - The colorbar does not allow distinguishing regions that have years of emergence in 1960 or in 2088.

**AC11:**
We have removed the first color, which was not used. All figures in the manuscript and the supplement that share the same colormap have been adjusted. All colors are unique now. Colors for very high and very low values are still both dark – though different. The gradual spatial change in the maps provide additional information whether a value is at the low or high end and should suffice to make clear whether the values are high or low.

Referee 2:

**RC:**
This is an interesting paper which presents a new approach for detecting emergence of climate change, applied to a basin in eastern Siberia. The approach looks justified and offers some important advantages over more conventional approaches. I have attached my marked copy of the manuscript with several grammatical corrections, but otherwise should be accepted.

**AC:** We have incorporated all suggested changes into the updated manuscript.

References

[revised manuscript text omitted]

Formatted Table